# Latent Planning via Expansive Tree Search

**Robert Gieselmann**
KTH Royal Institute of Technology
Stockholm, Sweden
robgie@kth.se

**Florian T. Pokorny**
KTH Royal Institute of Technology
Stockholm, Sweden
fpokorny@kth.se

## Abstract

Planning enables autonomous agents to solve complex decision-making problems by evaluating predictions of the future. However, classical planning algorithms often become infeasible in real-world settings where state spaces are high-dimensional and transition dynamics unknown. The idea behind latent planning is to simplify the decision-making task by mapping it to a lower-dimensional embedding space. Common latent planning strategies are based on trajectory optimization techniques such as shooting or collocation, which are prone to failure in long-horizon and highly non-convex settings. In this work, we study long-horizon goal-reaching scenarios from visual inputs and formulate latent planning as an explorative tree search. Inspired by classical sampling-based motion planning algorithms, we design a method which iteratively grows and optimizes a tree representation of visited areas of the latent space. To encourage fast exploration, the sampling of new states is biased towards sparsely represented regions within the estimated data support. Our method, called Expansive Latent Space Trees (ELAST), relies on self-supervised training via contrastive learning to obtain (a) a latent state representation and (b) a latent transition density model. We embed ELAST into a model-predictive control scheme and demonstrate significant performance improvements compared to existing baselines given challenging visual control tasks in simulation, including the navigation for a deformable object.

## 1 Introduction

To perform challenging control tasks in real-world environments, intelligent agents must reason across many temporal steps, often relying on high-dimensional sensor data such as images. Over the past decade, machine learning has significantly improved the state-of-the-art in image-based robotics and perception [34, 9, 2, 36, 17, 15]. However, existing methods are often limited to problems with relatively short time horizons and fail when the target is too far in the future. Reinforcement learning (RL), for example, is prone to failure in such scenarios due to sparse reward feedback and the resulting complexity of credit assignment [37, 16]. Classical planning algorithms [30], on the other hand, excel at solving temporally-extended decision problems, but typically require compact representations, state distance metrics and perfect knowledge of transition dynamics.

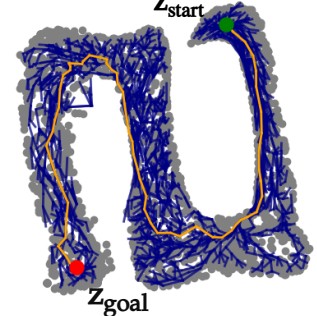

**Figure 1:** Sampling-based latent exploration with ELAST. A tree (blue) is iteratively grown and optimized while being bound to the estimated latent support region (gray).

Recent work proposes a mixture of planning and deep learning methods to enable planning from complex visual observations. Current approaches can be broadly divided into planning (a) directly in the observation space [44, 8, 35] and (b) in learned latent

36th Conference on Neural Information Processing Systems (NeurIPS 2022).

spaces [14] [43]. The first variant usually relies on generative modeling of video sequences to synthesize feasible image paths. The inherently high computational cost of training and deploying such generative models, as well as the burden to make visually accurate predictions, present key limitations of this type of method. Latent planning, on the other hand, maps the control problem to a lower-dimensional embedding space in order to reduce the complexity of the search space and facilitate the approximation of the dynamics. Moreover, this bypasses the computational bottleneck of synthesizing high-dimensional observations allowing fast computation of paths even in long-horizon settings.

A large number of existing latent planning strategies are based on trajectory optimization tools that originated in the optimal control literature [25]. In particular, gradient-free shooting-based strategies such as the Cross-Entropy Method (CEM) [3] or Model-Predictive Path Integral control (MPPI) [57] attracted much attention due to various applications in model-based RL [14] [39]. Numerical trajectory optimization through direct collocation [25] was recently used in [43] for planning over latent spaces. Despite their recent popularity, the aforementioned techniques are known to suffer from local minima when the optimization landscape is highly non-convex [51] which makes them less suitable for planning in geometrically complex spaces.

In this work, we pursue the intriguing idea of planning in latent spaces and focus on improving existing techniques by formulating planning as a search within the continuous latent state space. We present **E**xpansive **LA**tent **S**pace **T**rees (ELAST) which solves high-dimensional goal-reaching tasks through an explorative search within the latent data support. Our method differs from existing approaches in that it grows and optimizes a tree representation of previously visited areas, which can be leveraged to improve the efficiency of the exploration. This concept is largely inspired by single-query sampling-based planners [30] in the field of robot motion planning, in particular the asymptotically optimal version of Rapidly-exploring Random Trees (RRT) [31, 24] and Expansive Space Trees (EST) [18]. Motion planners are typically used for navigating robots through geometrically complex environments, analogously our motivation is to introduce similar concepts in latent spaces. It should be noted that direct application of these methods to our setting is **not possible because state metrics, sampling distribution, collision checks and other important quantities are difficult to define for learned embeddings**. To overcome these challenges, we use self-supervised contrastive learning to obtain a state representation which locally preserves certain geometric properties that favor planning, and in a second step approximate latent dynamics and state connectivity models from random interaction data. We demonstrate that our learned embedding allows us to effectively use noise-contrastive estimation (NCE) [12, 13] for the approximation of conditional transition densities.

Our main contribution is the presentation of ELAST, a new latent planning algorithm for long-horizon goal-reaching. In the scope of this work, we focus on visual input data represented by sequences of image observations. To the best of our knowledge, we are the first to present explorative tree search in latent spaces without the need of expert data or privileged information about the environment. Unlike static map-based approaches [44] [8] [10], our planner is compatible with parameterized settings, e.g. environments which are conditioned on the position of obstacles. Our evaluation on challenging visual control tasks in simulation shows significant performance improvements in downstream task success rate and path quality compared to existing baseline solutions.

## 2   Related Work

**Planning in Images Spaces**   A large body of prior works learn video dynamics to solve visual planning tasks [9, 7, 59, 40, 53]. Visual Foresight [7], for example, generates predictions of future frames to optimize next-step actions through shooting-based optimization. [53] extends this idea with offline Q-learning to obtain improved cost-to-go metrics in long-horizon settings. [40] uses a hierarchical architecture to produce subgoal images for temporally-extended navigation tasks. The approaches in [8, 19, 29] learn a distance metric using value-based reinforcement learning to construct a graphical representation of the observation space. Paths are then generated by solving the shortest path problem on the graph. In a similar fashion, [44] uses self-supervised training of a connectivity classifier to construct a map of neighboring image states. [35] builds upon this idea but uses an energy-based model to approximate the traversability between states. The methods presented in [27, 55] use a map-free approach which plans visual paths by predicting intermediate frames using a generative adversarial network [11]. In this work, we take a fundamentally different approach to

visual planning by instead solving the decision-making task in a learned embedding space, thereby bypassing the need for computationally demanding generative video models.

**Latent Planning and Control**    The idea of latent planning and control is to use compact representations which simplify decision-making and dynamics learning. [56] train a linear dynamics model to achieve locally-linear control from images in a learned embedding. [33, 48, 42] investigate contrastive-predictive coding to learn useful representations for control. [14] uses a reconstruction-based visual embedding and plans actions using shooting-based trajectory optimization via the Cross-Entropy Method (CEM) [3]. Similary, [39] uses Model-Predictive Path Integral Control (MPPI) [57] with learned dynamics models for dexterous robot manipulation tasks. [43] recently introduced collocation-based trajectory optimization [25] to model-based reinforcement learning in latent spaces. The methods in [50, 61] approach visual control tasks via gradient-based planning in a learned embedding space. [41] plans sequences of coarse subgoals by solving the shortest-path problem through direct optimization on the latent manifold. [22] implement an RRT-like [30] sampling-based planner within a learned embedding. The presented method is suitable for robot motion planning from image inputs and relies on the existence of state samplers and collision checking mechanisms. Similarly, [23] presents a task-conditioned sampling-based motion planner for efficient exploration in latent spaces but requires access to a predefined task space. The line of work in [45, 46] combines model-based RL with Monte Carlo Tree Search (MCTS) [5] to improve the capabilities in long-horizon decision-making tasks such as playing Atari games from video input. While limited to discrete action spaces, [20] recently extended this idea to continuous action spaces introducing a sampling-based policy iteration framework.

Similar to the works in [22] and [23], our method is highly inspired by classical sampling-based robot motion planners. However, in contrast, we do not require expert data, collision oracles, or other privileged information about the environment. ELAST is a single-query approach, meaning that it generates a new search tree for each planning query. Unlike map- or graph-based approaches [8, 19, 29, 44], this provides more flexibility by allowing planning beyond static environments, e.g., parameterizing on obstacle configurations. Compared to sampling-based approaches such as CEM, MPPI or continuous MCTS, we exploit the properties of our state embedding for keeping track of already visited areas, thereby encouraging fast exploration.

## 3  Preliminaries

**Sampling-based Motion Planning**    In the motion planning literature, the configuration space is defined as the space of all possible robot configurations, i.e., the specifications that accurately describe the state of a robot system [30]. Sampling-based motion planners solve continuous robot motion planning tasks by performing a search that probes the configuration space through sampling [30]. A prominent representative of sampling-based algorithms are Rapidly-exploring Random Trees (RRT) [31]. It grows a search tree by iteratively adding new samples drawn from the configuration space. At each step, a new state is connected to its nearest neighbor in the tree if the corresponding transition is valid. Instead of sampling directly from the state space, Expansive Space Trees (EST) [18] build a search tree by systematically expanding nodes in sparsely represented regions of the configuration space. The work in [24] presents an adaptation of sampling-based methods such as RRT which produce asymptotically optimal paths with respect to the number of planner iterations. Classical motion planners operate on well-described configuration spaces, e.g. robot joint space, and assume access to collision detection and state samplers. In this work, we bridge the gap between sampling-based motion planning and learning control from visual input.

**Noise-Contrastive Estimation**    Density estimation is a ubiquitous task in machine learning and statistics. Unnormalized density models, often known as energy-based models, offer flexible parameterizations for instance using expressive neural network architectures. A recent introduction to energy-based model optimization is given in [49]. Noise-contrastive estimation (NCE) [12, 13] is a widely-used training method for energy-based models due to its balance between statistical efficiency and computation. NCE trains a density model by distinguishing between points $x$ from the original data distribution $p_d$ and samples drawn from an auxiliary noise distribution $p_n$. The corresponding maximum likelihood objective, shown in Eq. 2, resembles nonlinear logistic regression. NCE simultaneously estimates an unnormalized model $p_m^0$ and normalization constant $b$, which allows to retrieve normalized density estimates $p_m$, where $\ln p_m(u) = p_m^0(u) + b$. Note that $\mathbb{E}$ is in practice

computed as an empirical mean from the training data.

$$m(u) = \frac{1}{1 + \nu \exp(-G(u))} \quad \text{with} \quad G(u) = \ln p_m(u) - \ln p_n(u) \tag{1}$$

$$\mathcal{J}_{\text{NCE}} = \underset{x \sim p_d}{\mathbb{E}} \big[ \ln[m(x)] \big] + \nu \underset{y \sim p_y}{\mathbb{E}} \big[ \ln[1 - m(y)] \big] \tag{2}$$

To ensure good approximation results, it is crucial that the support of $p_n$ contains $p_d$ without deviating too much from it. Moreover, $p_n$ should be computationally efficient to evaluate for any input. A central problem in NCE is therefore to find a proper choice for $p_n$ with respect to the unknown data distributions $p_d$. For detailed derivations, we refer to [12, 13].

**Contrastive Predictive Representations** Contrastive representation learning compares among data samples to learn embeddings which map similar data samples close together and dissimilar ones far apart [32]. Contrastive Predictive Coding (CPC) [54] uses this principle to learn latent representations $\mathcal{Z}$ for sequential data $\mathcal{X}$. The InfoNCE objective (Eq. 3) was derived which attempts to maximally preserve the mutual information between the $k$-step future observations $x_{t+k}$ and context representations $y_t$ which pool prior information $z_{\leq t}$ in the latent space $\mathcal{Z}$. Inspired by NCE, it identifies correlated pairs drawn from $p(x_{t+k}|y_t)$ among uncorrelated ones, given a user-defined similarity $f(x_{t+k}, y_t)$. [54] use $f(x_{t+k}, y_t) = \exp(z_{t+k}^T W_k y_t)$ which computes similarity by taking the dot product between the latent encodings $z_{t+k}$ and their corresponding future linear predictions $W_k^T y_t$.

$$\mathcal{L}_{\text{InfoNCE}} = -\underset{\mathcal{X}}{\mathbb{E}} \left[ \log \frac{f(x_{t+k}, y_t)}{\sum_j f(x_j, y_t)} \right] \tag{3}$$

CPC was shown to learn smooth representations useful for control in various different settings [48, 33, 58, 35]. We utilize an adaptation of it (Sec. 4.1) to recover embeddings that are compatible with our planning algorithm.

## 4 Expansive Latent Space Trees

We study goal-conditioned reaching tasks from visual inputs modeled using Markov Decision Processes (MDPs). The state space $\mathcal{S}$ is composed of image sequences $\mathcal{S} = \mathbb{R}^{N \times C \times W \times H}$, where N denotes the number of frames, C the number of channels, W the width and H the height of single frames. Moreover, we define the continuous action space $\mathcal{A} = \mathbb{R}^{d_\mathcal{A}}$ and the set of goal states $\mathcal{G} = \mathcal{S}$. In addition to single environments, we also consider families of environments conditioned on raw context observations $c \in \mathcal{C}$ (e.g. images) capturing task-relevant information, e.g. the position of obstacles. Instead of learning a controller directly from images, we instead seek a method which solves goal-reaching tasks in a lower-dimensional latent state space $\mathcal{Z} = \mathbb{R}^{d_\mathcal{Z}}$. In this regard, we assume no access to the underlying MDP dynamics, state distance metrics and only provide a fixed-size training dataset composed of random environment interactions.

This section presents ELAST, a new method for solving the described type of control problems through explorative tree search and local control within $\mathcal{Z}$. A schematic overview of our approach is presented Fig. 2. It consists of a state encoder $\phi$ (Sec. 4.1), transition density model $\psi$ (Sec. 4.2), forward dynamics model $h_f$ (Sec. 4.3) and a local policy $\pi$ (Sec. 4.4), each modelled using feedforward neural networks. During execution time, $\phi$ encodes the current and goal states $s_{\text{start}}, s_{\text{goal}} \in \mathcal{S}$ into the latent vectors $z_{\text{start}}, z_{\text{goal}} \in \mathcal{Z}$ (Fig. 2a). For context-conditioned tasks, we additionally feed a context-vector $c$ into the network. The planning module (Fig. 2d) then builds a search tree within $\mathcal{Z}$ that finds a sequence of latent states connecting $z_{\text{start}}$ to $z_{\text{goal}}$. To improve the quality of solutions in terms of temporal path length, the search tree is optimized during planning. This is done by rewiring the tree after every expansion using the transition density estimator $\psi$ (Fig. 2b). The planner then passes the computed discrete n-step solution path to the MPC controller module (Fig. 2c) which queues the corresponding intermediate states to be achieved by $\pi$.

### 4.1 Geometric Properties of the Encoder

We define an encoder $\phi : \mathcal{S} \times \mathcal{C} \to \mathcal{Z}$ which maps states $s_t$ conditioned on contexts $c$ into a latent space $\mathcal{Z} = \mathbb{R}^{d_\mathcal{Z}}$ with $d_\mathcal{Z} \ll N \times C \times W \times H$. For brevity, we for now focus on environments

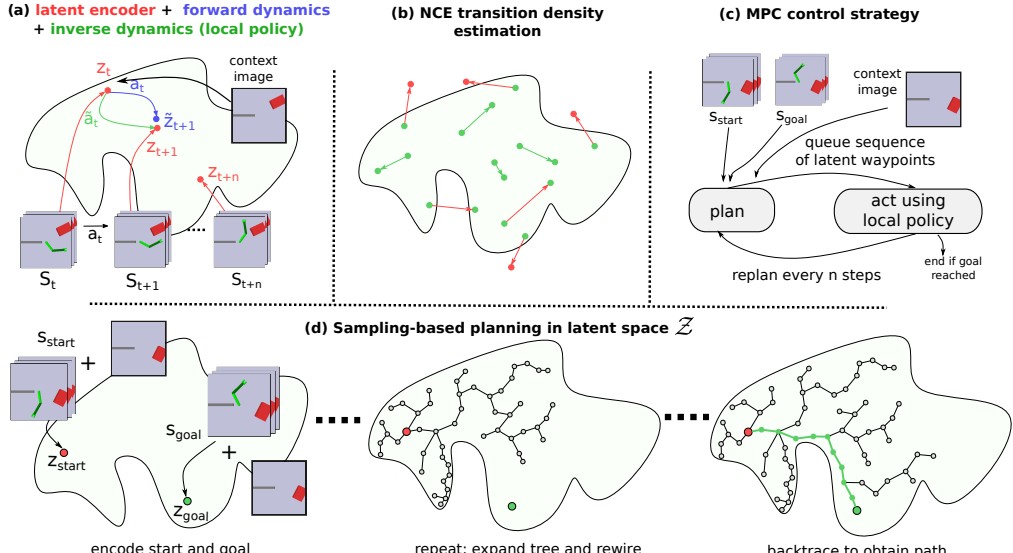

**Figure 2:** Overview of the different modules of ELAST. (a) training of latent encoder, dynamics and policy, (b) noise-contrastive estimation of transition densities $p(z_{t+1}|z_t, c)$, (c) model-predictive control framework, (d) planning via explorative tree search in $\mathcal{Z}$.

without a task-specific context, hence $\phi : \mathcal{S} \rightarrow \mathcal{Z}$. As further described in Sec. 4.3, our planner performs nearest-neighbor lookups to sample new nodes and rewire the tree. To enable simple and efficient neighborhood computations, a desirable property of $\phi$ is the preservation of the proximity of temporally subsequent states within $\mathcal{Z}$. This can be achieved by using a global state distance metric to enforce strict correspondence of distances during training. However, it typically requires privileged knowledge of the underlying MDP dynamics to define such a metric and is therefore not possible in our setting. Moreover, distances cannot be inferred directly from our training data which was collected by a sub-optimal policy. Instead, we rely on the knowledge of continuity between observations to enforce proximity of temporally adjacent states. More specifically, we employ contrastive learning via CPC (Sec. 3) and consider positive pairs $(z_{t+k}, \widetilde{z}_{t+k})$ where $\widetilde{z}_{t+k}$ denotes the predicted future latent state given the current state $z_t$ and sequence of actions $a_{t:t+k-1}$. To obtain $\widetilde{z}_{t+k}$, we simultaneously train an auto-regressive dynamics model $h_f : \mathcal{Z} \times \mathcal{A} \rightarrow \mathcal{Z}$. A similarity based on the squared Euclidean distance is used, hence $f(s_{t+k}, s_t) = e^{-\|\widetilde{z}_{t+k} - z_{t+k}\|_2^2}$. The encoder loss $\mathcal{L}_\phi$ is depicted in Eq. 4. In accordance with the observations in [48, 33, 58] we add a MSE loss on the predictions of $h_f$ (weighted by $K_h$) to encourage $h_f$ to be consistent with the true latent dynamics.

$$\mathcal{L}_\phi = - \underset{\mathcal{S}}{\mathbb{E}}\left[\log \frac{f(s_{t+k}, s_t)}{\sum_{s_j \in \mathcal{S}} f(s_j, s_t)}\right] + K_h \cdot \underset{\mathcal{Z}}{\mathbb{E}}\left[\sum_{i=0}^{k-1}(z_{t+1+i} - h_f(\widetilde{z}_{t+i}, a_{t+i}))^2\right] \qquad (4)$$

Optimizing $\phi$ with Eq. 4, we intend to achieve locality of correlated states while shaping dynamics that are predictable by a model $h_f$. For context-conditioned tasks, we additionally feed $c$ into the dynamics model, i.e. $h_f : \mathcal{Z} \times \mathcal{A} \times \mathcal{C} \rightarrow \mathcal{Z}$ and optimize $\phi$ considering the empirical expectation of the loss in Eq. 4 w.r.t. $\mathcal{C}$ (full objective in App. A.1).

## 4.2 Conditional Transition Density

Due to the symmetry of the critic $f$ in CPC (Sec. 4.1), the learned embedding does not encode the full causal relationship between states. Subsequent states are organized close together but we lose the information in which direction transitions were experienced, i.e the arrow of time. A model of the state connectivity is yet required for rewiring tree nodes and becomes indispensable for environments with asymmetric dynamics. To recover this knowledge, we approximate the conditional density $p(z_{t+1}|z_t, c)$ of latent transitions $z_{t:t+1|c}$ using noise-contrastive estimation (NCE) (Sec. 3). More specifically, we employ a parametric model $\psi : \mathcal{Z} \times \mathcal{Z} \times \mathcal{C} \rightarrow \mathbb{R}$ to represent the set of log density estimators given by the conditioning on the transition source node $z_t$ and environment context $c$. We

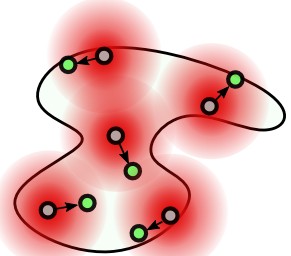

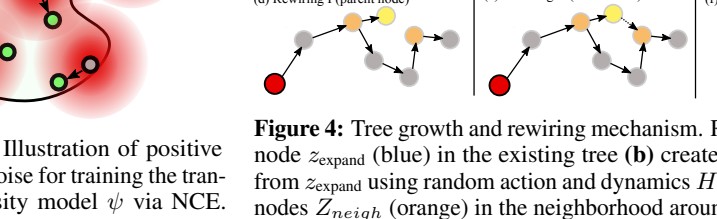

**Figure 3:** Illustration of positive pairs and noise for training the transition density model $\psi$ via NCE. Gaussian noise distribution $p_n$ in red, start states of transitions $z_t$ in gray and end points of transitions $z_{t+1}$ in green.

**Figure 4:** Tree growth and rewiring mechanism. Repeat: **(a)** select random node $z_{\text{expand}}$ (blue) in the existing tree **(b)** create new node $z_{\text{new}}$ (yellow) from $z_{\text{expand}}$ using random action and dynamics $H_f$ **(c)** find set of reachable nodes $Z_{neigh}$ (orange) in the neighborhood around $z_{\text{new}}$ **(d)** change parent of $z_{new}$ if there exist a node in $Z_{neigh}$ for which $z_{new}$ is reachable and which reduces the cost of reaching $z_{new}$ **(e)**, **(f)** alter parent for every nodes in $Z_{neigh}$ if transitioning from $z_{new}$ is possible and reduces traveling cost.

train $\psi$ by maximizing the expectation of the NCE objectives $\mathcal{J}_{\text{NCE}}$ over the empirical distribution of all conditionings for the observed data, hence $\mathcal{J}_\psi = \mathbb{E}_{c \sim \mathcal{C}, z_t \sim \mathcal{Z}} \left[ \mathcal{J}_{\text{NCE}}(c, z_t) \right]$. A discrete version of this strategy has previously been used in [38] for estimating the context-dependent probability of words in probabilistic language models.

The crux of NCE is to find a noise distribution $p_n$ which tightly encloses the support of the unknown target distribution (Sec. 3), in our case $p(z_{t+1}|z_t, c)$. Our key insight is however that due to the contrastive training of $\phi$, subsequent states are encouraged to be mapped close together in $\mathcal{Z}$. Hence, a simple choice for $p_n$ is found by centering a multivariate Gaussian $\mathcal{N}(z_t, \Sigma)$ at each starting state $z_t$ of a transition and setting the value of $\Sigma$ empirically given the absolute lengths of all observed latent transitions in the data (Fig. 3).

One motivation of using NCE instead of other density estimation techniques is that it allows the use of powerful neural network function approximators. Furthermore, the discriminative training procedure using $p_n$ provides examples of outlier points for which the model assigns low density. Extrapolating densities is an important feature which helps us to identify unlikely transitions outside the data support.

### 4.3 Tree Expansion and Optimization

Due to the unboundedness of the Euclidean embedding space, erroneous transitions during the tree expansion might result in the exploration of areas outside the support of the latent distribution. To reduce the number of detrimental transitions, we train a powerful dynamics ensemble estimator $H_f = \{h_f^1, .., h_f^m\}$ after the encoder training. Moreover, this allows us to evaluate the ensemble's predictive uncertainty [28] and reject highly uncertain predictions during planning.

Given the encoder $\phi$, forward dynamics $H_f$ and transition density model $\psi$, all the building blocks that compose our planner are introduced. For every new planning query, we start by adding $z_{\text{start}}$ to the tree and then alternate between expanding and optimizing the search tree. Fig. 4 illustrates the mechanisms behind this procedure.

**Tree Expansion**  During the expansion step, we sample a node $z_{\text{expand}}$ from the current tree and generate a new state $z_{\text{new}}$ from it using $H_f$ and a random action $a \sim \mathcal{A}$. The newly generated transition is discarded if the density estimated by $\psi$ is below a threshold $\tau_t$ or the predictive uncertainty $\text{Var}(\{h_f^i, i = 1..m\})$ exceeds a threshold $\tau_e$. If both tests are passed, we connect $z_{\text{new}}$ to the tree and continue with the rewiring step.

**Rewiring**  We optimize the search tree by rewiring edges locally around each newly added node if this reduces the traveling distance from the root. Firstly, we find the set of neighboring tree nodes $Z_{neigh}$ within a ball of radius $r_{\text{neigh}}$ around $z_{\text{new}}$. Next, we select the elements in $Z_{neigh}$ that are reachable from $z_{\text{new}}$, i.e. the associated transition density $\geq \tau_t$. For all the remaining nodes, including $z_{\text{expand}}$, we determine the number of steps to travel from $z_{\text{start}}$ to $z_{\text{new}}$. The node associated with the

minimum cost becomes the new parent node of $z_{\text{new}}$. For each node in $Z_{neigh}$, we then make $z_{\text{new}}$ the parent node if such a transition is likely and it reduces the traveling cost from $z_{\text{start}}$. Essentially, rewiring reduces travel costs from the root by looking for lower cost paths through existing nodes in the tree. Note that similar techniques are used within asymptotically-optimal sampling-based motion planners, e.g. RRT* [24]. The key difference, however, is that our planner operates in a learned state embedding with approximated latent dynamics and connectivity. Further, note that we consider the path cost as the number of steps in time, not the geometric length of the piecewise linear latent path.

The planning stops after $n_{\text{iter}}$ expansion steps. Solution paths are then determined by backtracking up the tree. We return the path with the lowest travel cost among the paths that end near $z_{\text{goal}}$ (inside a sphere of radius $r_{\text{goal}}$). If there are no such paths, we simply take the one with the smallest remaining Euclidean distance to $z_{\text{goal}}$.

**Node Sampling Biases**    Uniform sampling of $z_{\text{expand}}$ leads to slow growth of the search tree with respect to the covered space. To encourage the planner to quickly grow into unexplored regions, we need to bias the sampling towards regions of $\mathcal{Z}$ that are less represented in the current tree. With probability $p_{\text{sparse}}$, we therefore sample $z_{\text{expand}}$ using a distribution weighted by the inverse number of reachable neighbors for each node. The random uniform node selection is continued with probability $p_{\text{uniform}}$. We also use $p_{\text{goal}}$ to select the node closest to $z_{\text{goal}}$ in terms of Euclidean latent distance. We found that this simple heuristic promotes sufficient exploration towards the goal. For more details on sampling biases, see App. A.1.2.

## 4.4   Integration with Model-Predictive Control

In order to use ELAST for goal-reaching control, we integrate it into a simple Model Predictive Control (MPC) scheme (Fig. 2c). The controller queues the planned sequence of latent states as waypoints and replans after $n_{\text{replan}}$ interactions. Between planning, the local control policy $\pi : \mathcal{Z} \times \mathcal{Z} \times \mathcal{C} \to \mathcal{A}$ navigates to the next waypoint. For simplicity, we employ a simple one-step policy which is trained using the supervised loss $\mathcal{L}_\pi = \mathbb{E}_{\mathcal{Z}} \left[ (a_t - \pi(z_t, z_{t+1}, c))^2 \right]$.

# 5   Experiments

In this section, we evaluate the effectiveness of ELAST for solving challenging downstream reaching tasks from visual observations. For that purpose, we use the closed loop control setting described in Sec. 4.4 and measure the performance in terms of average success rate. Moreover, we investigate the impact of different components and hyperparameters of our method on the quality of the computed solutions.

## 5.1   Experimental Setup

**Baselines**    To put ELAST in relation to existing work, we use the following baselines. *PlaNet* [14], a latent planning algorithm based on shooting-based trajectory optimization via the Cross-Entropy Method (CEM) [3]. *Hallucinative Topological Memory* (*HTM*), an image-based planning method for context-conditioned reaching tasks. *HTM* plans paths on a map of raw images generated by sampling observations from a Variational Autoencoder [26]. To decouple the effect of representation learning and planning, we evaluate several methods that build on our learned CPC embedding. *CPC-CEM* uses planning via CEM using the Euclidean latent distance towards the goal as cost. Similarly, *CPC-Coll.* uses trajectory optimization through direct collocation. *CPC-GCBC* presents a behavioral cloning method trained on CPC latent states. Finally, we implemented a goal-conditioned behavior cloning policy *V-GCBC* which is trained directly from images. In addition, we tested an offline version of the method presented in [43], but it did not produce sensible results under our experimental conditions and is therefore not included in the following evaluation (see App. C.5). Detailed information on the hyperparameters and the implementation of the baselines can be found in App. C.

**Control Tasks**    To test our method for specific environmental characteristics, we designed the toy environments shown in Fig. 5. In *BlockS*, a block agent must navigate through an S-shaped corridor representing a long-horizon and a geometrically non-convex setting. *BlockAsym* introduces asymmetric transition dynamics through a unidirectional stream on one side of the workspace. Parameterized environments that depend on the position of obstacles are explored in both *BlockParam*

**Table 1:** Success rates (%) for test scenarios averaged over three independent training runs

| METHOD | BLOCKS | | BLOCK ASYM | BLOCK PARAM | PLANAR ARM | REACH | BUTTON | DRAWER | HAMMER | CABLE | CABLE PARAM |
|---|---|---|---|---|---|---|---|---|---|---|---|
| | MEDIUM | HARD | | | | | | | | | |
| ELAST | $99 \pm 1$ | $96 \pm 3$ | $100 \pm 0$ | $84 \pm 2$ | $88 \pm 1$ | $96 \pm 2$ | $88 \pm 5$ | $84 \pm 12$ | $22 \pm 2$ | $95 \pm 2$ | $79 \pm 11$ |
| CPC-CEM | $53 \pm 2$ | $9 \pm 4$ | $21 \pm 7$ | $68 \pm 3$ | $64 \pm 1$ | $68 \pm 6$ | $1 \pm 0$ | $32 \pm 2$ | $0 \pm 0$ | $29 \pm 4$ | $17 \pm 3$ |
| CPC-COLL. | $15 \pm 1$ | $0 \pm 0$ | $7 \pm 5$ | $3 \pm 2$ | $21 \pm 7$ | $3 \pm 2$ | $1 \pm 1$ | $44 \pm 2$ | $0 \pm 0$ | $5 \pm 1$ | $26 \pm 1$ |
| CPC-GCBC | $20 \pm 2$ | $0 \pm 0$ | $0 \pm 0$ | $29 \pm 4$ | $57 \pm 2$ | $31 \pm 2$ | $20 \pm 14$ | $2 \pm 1$ | $0 \pm 0$ | $26 \pm 5$ | $20 \pm 1$ |
| PLANET | $17 \pm 1$ | $0 \pm 0$ | $0 \pm 0$ | $11 \pm 0$ | $42 \pm 12$ | $6 \pm 1$ | $18 \pm 3$ | $34 \pm 9$ | $0 \pm 0$ | $5 \pm 1$ | $38 \pm 5$ |
| HTM | $7 \pm 3$ | $0 \pm 0$ | $15 \pm 8$ | $28 \pm 11$ | $43 \pm 7$ | $5 \pm 6$ | $45 \pm 4$ | $3 \pm 2$ | $0 \pm 0$ | $19 \pm 3$ | $30 \pm 2$ |
| V-GCBC | $25 \pm 8$ | $0 \pm 0$ | $0 \pm 0$ | $35 \pm 2$ | $56 \pm 1$ | $37 \pm 1$ | $81 \pm 9$ | $48 \pm 3$ | $13 \pm 2$ | $1 \pm 1$ | $1 \pm 1$ |
| RANDOM | $8 \pm 2$ | $0 \pm 0$ | $0 \pm 0$ | $1 \pm 1$ | $2 \pm 1$ | $0 \pm 0$ | $1 \pm 0$ | $4 \pm 2$ | $0 \pm 0$ | $8 \pm 3$ | $1 \pm 1$ |

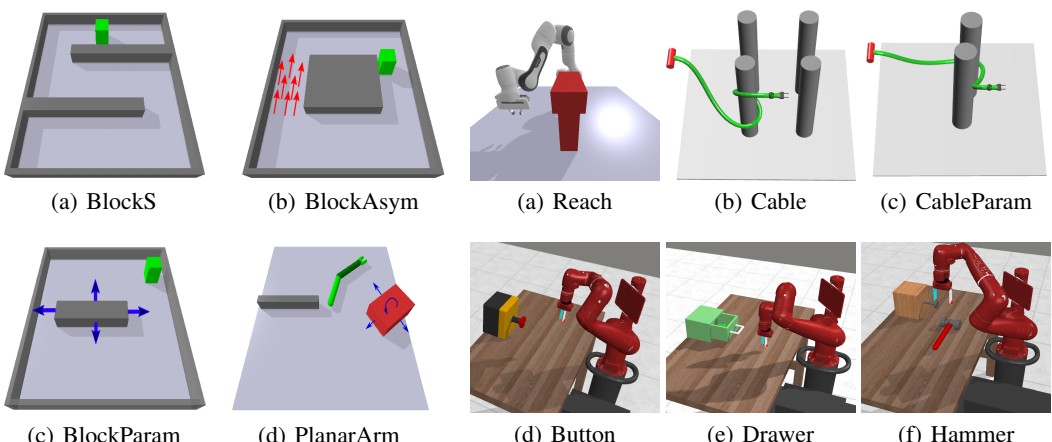

(a) BlockS   (b) BlockAsym

(a) Reach   (b) Cable   (c) CableParam

(c) BlockParam   (d) PlanarArm

(d) Button   (e) Drawer   (f) Hammer

**Figure 5:** Planar control environments

**Figure 6:** Visually-complex robotics tasks

and *PlanarArm*. Additionally, we evaluated the visually demanding robot control tasks presented in Fig. 6. These include a robot reaching task (Fig. 6a) and a static and parameterized version of a cable navigation problem (Fig. 6b and c). In addition, we implemented three robot manipulation tasks (Fig. 6 d-f) adapted from the metaworld benchmark [60].

We form states $s_t$ by stacking three consecutive RGB frames of resolution $64 \times 64$. In parameterized environments, we obtain context observations $c$ by taking the latent encodings given by a convolutional autoencoder that was trained separately on raw context images. For all environments, we evaluate the performance in reaching distant goals $s_{\text{goal}}$ with respect to a task-specific maximum allowed number of interactions. We evaluate the success rates of trajectories in 100 unseen scenarios for each task. Detailed information about the implementation of the environments and the experimental evaluation can be found in App. A, B.

## 5.2 Experimental Results

**How does the performance of ELAST compare to other methods?** The results of our benchmark evaluation are shown in Table 1. Overall, we observe a significantly higher average success rate for MPC control with ELAST compared to the selected baselines. This is true for the planar environments as well as for the visually complex manipulation tasks. The ablation comparison with *CPC-CEM*, *CPC-Coll.* and *CPC-GCBC* supports that not only the choice of representation, but also our planning strategy contributes to the success of ELAST. Fig. 7 illustrates the exploration behavior in the *BlockS* environment for ELAST and *CPC-CEM*. It displays the Isomap embeddings [52] of the explored regions of the latent space with respect to different numbers of sampled states. It can be seen that our method expands faster and connects the goal to the tree with less than 7500 generated states. However, *CPC-CEM* did not explore deep enough to reach the goal, even given a much larger number of samples ($I{=}10$, $H{=}50$, $K{=}1000$ corresponds to $10^5$ generated states). *HTM* did not provide robust solutions compared to our approach. While the generated images were mostly

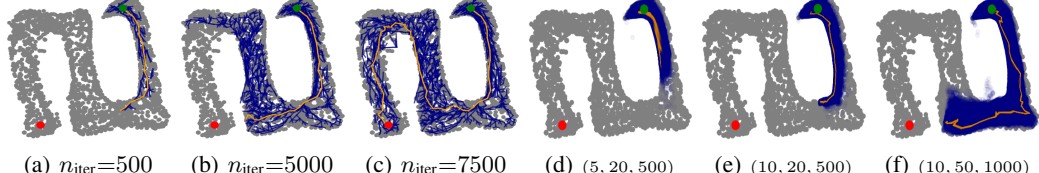

(a) $n_{\text{iter}}$=500    (b) $n_{\text{iter}}$=5000    (c) $n_{\text{iter}}$=7500    (d) $(5, 20, 500)$    (e) $(10, 20, 500)$    (f) $(10, 50, 1000)$

**Figure 7:** Exploration behavior of ELAST and CEM illustrated within 2D Isomap embeddings for the *BlockS* task. The goal (red) and start (green) lie on different ends of the corridor. (a)-(c): ELAST: search tree (blue) and best path (orange) at different iterations $n_{\text{iter}}$. (d)-(f): CEM: visited area (blue) and final population (orange) for different $(I, H, K)$ with iterations $I$, horizon $H$ and $K$ candidates.

visually sound, some remaining image artifacts resulted in shortcuts in the graph, often rendering entire paths infeasible. This result supports our motivation to pursue planning in a lower-dimensional space to avoid the challenge of obtaining accurate high-dimensional observations. While our method performed best among all baselines for the *Hammer* environment, we achieved only 22% success for this task. We attribute this to the difficulty of planning through the narrow passages introduced by the grasping subtask in this environment (see App. D.10). We also found that the agent often failed to grasp the object correctly, which might be improved in the future by using more sophisticated RL policies instead of the short-sighted policy $\pi$. Visualizations of successful latent paths for all environments can be found in App. D.9.

**How does our NCE transition density model impact the performance of the planner?** To answer this question, we tested ELAST in the *BlockS* and *Cable* environments without rejecting transition during planning. We observed a significant drop in performance for both tasks from $96 \pm 3\%$ to $51 \pm 4\%$ in *BlockS-hard*, respectively from $95 \pm 2\%$ to $25 \pm 5\%$ in the *Cable* task. Fig. 8a shows the predicted neighboring latent states with respect to a node in the unidirectional passage in the *BlockAsym* environment. As shown, our transition density model correctly determines state connectivity even under highly asymmetric transition dynamics. Without transition rejection enabled, the performance on this task dropped to $73 \pm 5\%$ as a result of infeasible shortcut paths against the direction of the stream. The results indicate that $\psi$ generates

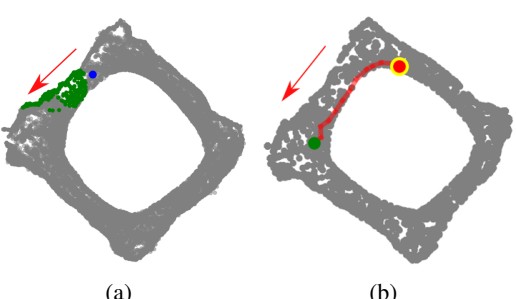

(a)      (b)

**Figure 8:** (a): Predicted neighboring states (green) for node (blue) within the unidirectional passage in *Block-Asym* (projected with Isomap). Obtained by thresholding estimated transition densities (b): Example of infeasible path from an initial state (green) towards a goal (red) that occurs when transition rejection is disabled

useful transition density estimates which is indeed crucial for the performance of our method.

**Does the rewiring affect the quality of generated paths?** The tree rewiring strategy (Sec. 4.3) encourages smooth and near-optimal paths which is important to finish the control tasks within the allowed contingent of environment steps. We evaluated ELAST on the *BlockS-hard* and the *Button* tasks without optimizing the tree. The success rates decreased drastically from $96 \pm 3\%$ to $40 \pm 4\%$ in *BlockS-hard* and from $88 \pm 5\%$ to $50 \pm 2\%$ in the *Button* environment. For the remaining successful episodes, we observed an increase of average trajectory length from $36 \pm 1$ to $93 \pm 2$ and from $38 \pm 6$ to $72 \pm 5$ steps in the *BlockS-hard*, respectively the *Button* task. Our results confirm the necessity of path rewiring to reduce trajectory lengths and thereby improve the success of our method.

**Importance of sparse node sampling bias** The exploration behavior of ELAST is largely determined by the weighting of the sampling biases. Fig. 9 illustrates the computed latent search trees for planning with $p_{\text{sparse}} = 0.78$ and $p_{\text{sparse}} = 0.0$ for the *BlockAsym* task. As shown, the tree does not reach the goal in the case of $p_{\text{sparse}} = 0.0$ which in fact led to a drop in performance to 0% on this task. Similar trends were observed for other environments (App. D.5). Our results suggest that sampling nodes in sparsely populated areas is very effective in speeding up the exploration of the search space.

For all experiments, we used $p_{uniform}$=0.2, $p_{sparse}$=0.78, and $p_{goal}$=0.02, which were found to be sufficient to achieve a balance between exploration and goal-directed node selection.

Depending on the difficulty of the environment, the number of planning iterations was chosen between $n_{iter}$=2500 and $n_{iter}$=10000. A detailed overview of planning hyperparameters is given in App. A.1.2. Moreover, we provide a collection of further ablation experiments in D. This include the impact of dynamics ensemble (App. D.3) and a comparison of using our planning on a $\beta$-VAE latent space instead of the CPC embedding (App. D.1).

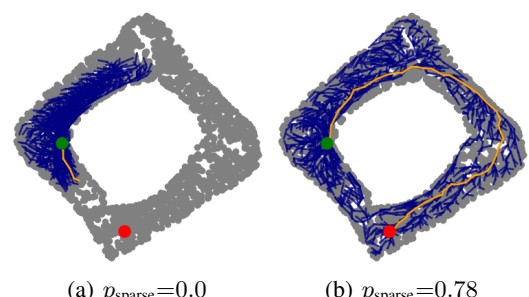

(a) $p_{sparse}$=0.0        (b) $p_{sparse}$=0.78

**Figure 9:** Impact of parameter $p_{sparse}$ on the tree exploration in the *BlockAsym* environment. Computed paths (orange) and search tree (blue) for $n_{iter} = 5000$, $p_{goal} = 0.02$ and $p_{uniform} = 1.0 - p_{goal} - p_{sparse}$ (visualized using Isomap)

## 6  Limitations and Future Work

The type of decision problems studied in this work is based on the assumption that the full state of the environment can be inferred from the information encoded in the image sequences. However, in robot manipulation tasks, partially observable environments are often encountered, e.g., due to visual occlusions or sensor uncertainties. To extend ELAST to partially observable MDPs (POMDPs), one could replace our deterministic encoder and integrate filtering techniques to update a belief over latent states. A similar approach was taken in [14], which uses a recurrent neural network architecture to integrate knowledge over time. In addition, the investigation of sampling-based belief space planning methods (e.g. [4]) in the context of latent spaces represents an interesting future direction.

In Sec. 4.3, we introduced the node sampling bias in order to quickly explore the latent space. Previous work employed deep generative networks to learn state sampling for classical robot motion planning [21]. Similarly, planner efficiency could be improved by incorporating learned sampling heuristics that consider task-specific information to promote the selection of promising nodes. Currently, ELAST performs random sampling of actions during tree expansion, which could be ineffective for control tasks with high-dimensional action spaces. In addition to node selection, our method could be improved by using heuristics to select actions.

In the context of this work, we focus on planning in goal-reaching scenarios with long time horizons. Similar to [14, 47, 43], it would be interesting to adapt our method to the RL setting. This would allow applications beyond reaching, e.g., achieving goals that are not defined by a single state. Instead of finding the shortest path to one particular latent state, the planner would need to search for a reward-maximizing trajectory.

## 7  Conclusion

We presented ELAST, a method for solving high-dimensional and long-term goal reaching through planning in latent spaces. Our approach combines ideas from classical sampling-based motion planning and self-supervised contrastive learning, and initiates a tree search bounded by the estimated support region of the latent space. The effectiveness of ELAST was demonstrated in challenging and visually complex control environments. We hope that our work will inspire new planning methods that incorporate both motion planning and machine learning concepts.

## Acknowledgements

This work was partially supported by the Wallenberg AI, Autonomous Systems and Software Program (WASP) funded by the Knut and Alice Wallenberg Foundation.

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
