# A  Expansive Latent Space Trees - Details and Implementation

Code repository available at `https://github.com/KRobG/expansive_latent_space_trees`.

## A.1  State Encoder for Context Environments

For environments with a task-specific context, we train the encoder $\phi : \mathcal{S} \times \mathcal{C} \to \mathcal{Z}$ with the loss $\mathcal{L}_{\phi,\text{context}}$ in Eq. 5 by computing the empirical expectations over context encodings $c \in \mathcal{C}$ with $f(s_{t+k}, s_t, c) = e^{-\left\| \tilde{z}_{t+k|c} - z_{t+k|c} \right\|_2^2}$.

$$
\mathcal{L}_\phi(c) = -\underset{\mathcal{S}}{\mathbb{E}} \left[ \log \frac{f(s_{t+k}, s_t, c)}{\sum_{s_j \in \mathcal{S}} f(s_j, s_t, c)} \right] +
$$

$$
K_h \cdot \underset{\mathcal{Z}}{\mathbb{E}} \left[ \sum_{i=0}^{k-1} (z_{t+1+i} - h_f(\tilde{z}_{t+i}, a_{t+i}, c))^2 \right] \tag{5}
$$

$$
\mathcal{L}_{\phi,\text{context}} = \underset{\mathcal{C}}{\mathbb{E}} \left[ \mathcal{L}_\phi(c) \right]
$$

### A.1.1  Network Architectures

We used the same network architectures and latent space sizes of 16 in all experiments. The encoder parameters are shown in Table 2. The density model $\psi$, single forward models $h_f$ and policy $\pi$ use four layers each consisting of 64 neurons and LeakyRelu activation functions in the hidden layers. For context conditioned environments, we additionally feed the context into the encoder by concatenating the input images.

**Table 2:** Hyperparameters of encoder $\phi$

| Parameter | Value |
|---|---|
| Filter | [16,16,16,32,64,64] |
| Kernels | [3,4,4,4,4,4] |
| Strides | [1,2,1,2,2,2] |
| Activation | LeakyRelu |
| Dense layers | [256,128] |
| Latent dimension | 16 |

In all experiments, $h_{\text{fwd}}$, $\psi$ and $\phi$ are represented using feed-forward neural networks with 64 neurons per layer, LeakyRelu activation functions and three hidden layers for unparameterized and 4 hidden layers for parameterized environments.

### A.1.2  Planning Module

The hyperparameters of our planning module are listed in Table 3. We use $n_{\text{iter}}$=500 in *Hammer*, 7500 in *Button*, *Reach* and 10000 in *BlockS* and *Drawer*. For all other tasks, we use 5000 planner iterations. The MPC controller replans after $n_{\text{replan}}$=50 steps in *Button*, *Drawer* and *BlockS*. Due to the accuracy required to correctly grasp the object in the *Hammer* environment, we replan after $n_{\text{replan}}$=3 for this task. For all remaining environments, we set $n_{\text{replan}}$=25. During exploration, nodes are chosen uniformly with probability $p_{\text{uniform}} = 0.2$. With probability $p_{\text{sparse}}$=0.78, the nodes are sampled weighted by the inverse number of neighboring nodes in the tree. With probability $p_{\text{goal}}$, we pick the node that lies closest to the goal (Euclidean distance). In order to keep the tree sparse, we discard newly generated latent states that are closer than $r_{\text{discard}}$ (Euclidean distance) to an existing node in the current tree.

### A.1.3  Training Details

For all tasks, we train $\phi$ for $5 \cdot 10^4$ iterations using batch sizes of 64 for unconditioned tasks. For training context-conditioned environments, using the loss in Eq. 5, we sample 64 transition tuples for 8 context vectors from the data. We train each dynamics model $h_f$ and policy $\pi$ for $5 \cdot 10^5$ iterations

**Table 3:** Hyperparameters of planning module

| Parameter | Value | Description |
|---|---|---|
| $n_{\text{iter}}$ | $\in \{500, 5000, 7500\}$ | Num. of sampled states (including rejections) |
| $n_{\text{replan}}$ | $\in \{3, 25, 50\}$ | Num. of steps until replanning |
| $p_{\text{uniform}}$ | 0.2 | Probability of sampling expansion node uniformly |
| $p_{\text{sparse}}$ | 0.78 | Probability of sampling expansion node sparsely |
| $p_{\text{goal}}$ | 0.02 | Probability of sampling node closest to goal |
| $r_{\text{neigh}}$ | 1.5 x avg. neigh. dist in $\mathcal{Z}$ | Radius to pre-select neighbors for rewiring |
| $r_{\text{discard}}$ | 0.5 x avg. neigh. dist in $\mathcal{Z}$ | Radius to discard sampled states |
| $\tau_{\text{t}}$ | $\in \{1, 2, 3, 5\}$ | Reject transition if density below $\tau_{\text{t}}$-th percentile of all transition densities in data. |
| $\tau_{\text{e}}$ | $\in \{95, 98, 99\}$ | Reject state/transition if predictive variance of transition is above $\tau_{\text{e}}$-th percentile of all predictive variances of latent transitions in data. |

each using batch sizes of 64. NCE training of the denstity model $\psi$ was done for $1.5 \cdot 10^5$ iterations using 64 negative samples drawn from the Gaussian noise distribution $p_n$ for each 64 inlier samples. For context-conditioned tasks, we instead use 32 inlier samples for 4 random contexts. The dynamics ensembles are composed of three networks trained independently with different seeds.

Training was performed on a GPU cluster. The total amount of compute for training all models (3 seeds) including baselines is estimated around 588 hours (wall clock and single GPU).

# B  Datasets and Environments

The environments *BlockS*, *BlockAsym*, *BlockParam*, *PlanarArm* and *Reach* were implemented in PyBullet [6]. The *Button*, *Drawer* and *Hammer* tasks were adapted from the environments in [43] which are based on the Metaworld benchmark [60] (simulated in Mujoco). We implemented *Cable* and *CableParam* in AGX Dynamics [1], a proprietary physics engine which provides specialized classes for the implementation of deformable objects. The sizes of the training datasets are described in Table 4 and Table 5.

**Table 4:** Dataset sizes for unparameterized environments

| Parameter | *BlockS/ BlockAsym* | *Panda/Button/Drawer/Hammer* | *Cable* |
|---|---|---|---|
| Number of trajectories | 1000 | 1000 | 2000 |
| Trajectory length | 25 | 50 | 25 |

**Table 5:** Dataset sizes for parameterized environments

| Parameter | *BlockParam/ PlanarArm* | *CableParam* |
|---|---|---|
| Number of contexts | 300 | 450 |
| Trajectories per context | 20 | 40 |
| Trajectory length | 25 | 20 |

**BlockS**  In this environment, the velocity of a block object is controlled in order to match a certain goal configuration within the S-shaped planar workspace. During evaluation, start and goal configurations are generated such that they are separated by one wall (*BlockS-medium*) or two walls (*BlockS-hard*).

**BlockAsym**  Similar to *BlockS*, yet, an unidirectional steam is introduced on the left side of the corridor. For testing start and goal states were sampled on the opposite sites of the stream such that the agent cannot pass the stream and therefore must navigate around the large obstacle in the center of the workspace.

**BlockParam**   Another version of the block environments which is parameterized by the position of the wall in the center of the workspace. For evaluation, we sample start and goal states on the opposite sides of the wall.

**PlanarArm**   The joints of a 2 DOF planar robot arm are actuated to steer it into a certain configuration. We generated test scenarios by sampling start and goal configurations with a minimal angular distance of $\pi/2$ between the rotational angles of the base joint.

**Reach**   The endeffector position of a 6-joint robot arm is controlled in order to achieve a certain goal configuration of the robot. For testing, we consider start and goal states which lie on opposite sides with respect to a T-shaped obstacle located in the middle of the workspace.

**Button, Drawer and Hammer**   We reused the sparse metaworld environments from [43]. Yet, only for the *Button* task, we increased the inital distance between endeffector and button to complicate the problem in terms of required planning horizon.

**Cable and CableParam**   The loose end of a deformable cable object is navigated around four (two in parameterized case) cylindrical obstacles to reach a certain goal configuration of the cable segments. During evaluation, start and goal segment configurations were generated by sampling two distinct states from a set of predefined configurations and executing 10-steps of random actions. To ensure sufficiently difficult planning scenarios in *CableParam*, we only consider situations in which the initial and goal positions of the cable ends are separated by a distance of at least 75mm. The x-y positions of the obstacles in *CableParam* are randomized.

In all experiments, we consider an episode successful if the goal configuration was reached within $n_{\max}$ environment steps. Table 6 presents the used values of $n_{\max}$ for all environments.

**Table 6:** Maximum number of steps for testing

| Environment | Max Steps |
| --- | --- |
| BlockS (medium) | 100 |
| BlockS (hard) | 100 |
| BlockAsym | 100 |
| BlockParam | 50 |
| PlanarRobot | 75 |
| PandaReach | 150 |
| Button (Metaworld) | 75 |
| Drawer (Metaworld) | 75 |
| Hammer (Metaworld) | 75 |
| Cable | 50 |
| CableParam | 50 |

## C   Baselines

### C.1   CPC-CEM

This baseline applies planning via CEM [3] using the same CPC embedding and dynamics ensemble $H_f$ used for ELAST. We compute trajectory cost by summing the per state Euclidean distances with respect to the goal latent encodings. In the *Hammer* environment, we replan after three environment interactions and for all other tasks after 25 time steps. The used CEM hyperparameters are presented in Table 7.

### C.2   CPC-Collocation

Inspired by the work in [43], we implemented a collocation-based trajectory optimization strategy in the latent space. Again, this baseline reuses the CPC embedding and dynamics ensemble $H_f$ from ELAST. Collocation optimizes a fixed-length sequences of states and actions considering multiple objectives such as dynamics feasibility, action feasibility, path cost and satisfaction of boundary

**Table 7:** Parameters of CEM Planner

| Parameter | Value |
| --- | --- |
| Horizon (H) | 50 |
| Candidates (K) | 1000 |
| Iterations (I) | 10 |
| Elite size | 100 |

values. For optimization, we use gradient-based optimization via the *Adam* optimizer for 2000 iterations given a trajectory resolution of 50. In general, the results of this method were highly sensitive to the correct choice of weights of the different terms in the optimization objective which we found to be difficult to tune in our settings.

## C.3 CPC-GCBC

This baseline represents a goal-conditioned behavioral cloning policy trained on top of the CPC latent embedding. The policy is represented by a feed-forward neural network consisting of four hidden layers with 64 neurons each and LeakyRelu activation functions. For training, we sample goal states from the set of future states within the same trajectory.

## C.4 PlaNet

We implemented PlaNet [14] based on the implementation from `https://github.com/zchuning/latco/`. We used the same network architecture provided in [14] but adapted PlaNet to the offline setting without allowing further environment interaction. For learning the reward function, we label batches of transitions during training given privileged access to the underlying goal condition of the environment. Planning was done with CEM with the hyperparameters provided in Table 7. We execute PlaNet and replan a new trajectory every three steps of interactions for the *Hammer* task and otherwise after 25 steps.

## C.5 LatCo

LatCo [43] uses collocation-based trajectory optimization on top of the PlaNet latent and dynamics model [14]. We reused the code from `https://github.com/zchuning/latco/` which we adapted to the goal-conditioned offline setting. Note that the original implementation uses a Levenberg-Marquardt optimizer which was at the time not available in the repository. Instead, we used gradient-based optimization via the *Adam* optimizer. We excluded this method from the numerical comparison in Table 1 since we were unable to produce robust trajectories with this method. As a reason, we suspect the difficulty in manually tuning the weighting of cost in the optimization objective as well as the sensitivity to the initialization.

## C.6 Hallucinative Topological Memory

We implement *HTM* [35] based on the implementation provided in `https://github.com/thanard/hallucinative-topological-memory` and used the same architectures for all networks and planner hyperparameters as the authors. Replanning is performed after 10 environment steps. During testing, we observed that the VAE generally produced visually appealing images. Yet, we found that the resulting maps contained invalid shortcut connections which frequently led to infeasible paths.

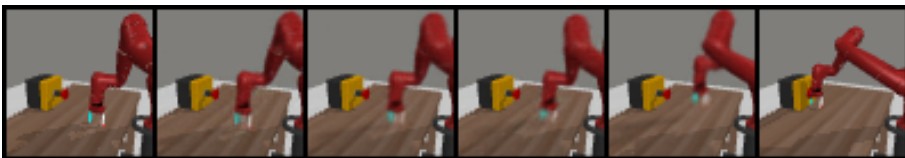

**Figure 10:** Example of invalid image sequence (left to right) planned with *HTM*. Sequence contains shortcut connection between the last two frames.

### C.7 Visual Goal-conditioned Behavioral Cloning

This baseline implements a goal-conditioned behavioral cloning policy trained directly from raw image observations. The policy modelled using a convolutional neural network with the same architecture as the ELAST encoder (see Table 2). Similar to *CPC-GCBC*, we sample goals from future states of the same trajectories during training.

# D Further Experimental Results and Ablations

### D.1 The impact of the contrastive embedding

To verify the necessity of using the CPC embedding (Sec. 4.1), we tested an ablated version of ELAST on a representation learned with a $\beta-$VAE [26] (latent dim. 16). For this comparison, we use the same architecture for all networks and only replace the type of loss in the encoder objective. Fig. 11 shows the Isomap embeddings of the latent state spaces for the *BlockS* environment obtained by CPC (b), $\beta-$VAE (c). We also tested a $\beta-$VAE model with enforced dynamics (d) for which we jointly optimized a forward dynamics model during encoder training. The visualization indicates that the CPC embedding better distinguishes between temporally close and distant states. Compared to ELAST with CPC, we observed a decrease of $90\%$ in success on the *BlockS-medium* when using a $\beta-$VAE. Training the dynamics model and the encoder together helped introducing structure into the $\beta-$VAE latent space. Nevertheless, using this version instead of CPC resulted in a decrease of $24\%$ for the *BlockS-medium* and $53\%$ for the *BlockS-hard* task. The results suggest that the CPC objective is better suited for our planning method, where it is critical that neighboring nodes are correctly identified during tree expansion and optimization.

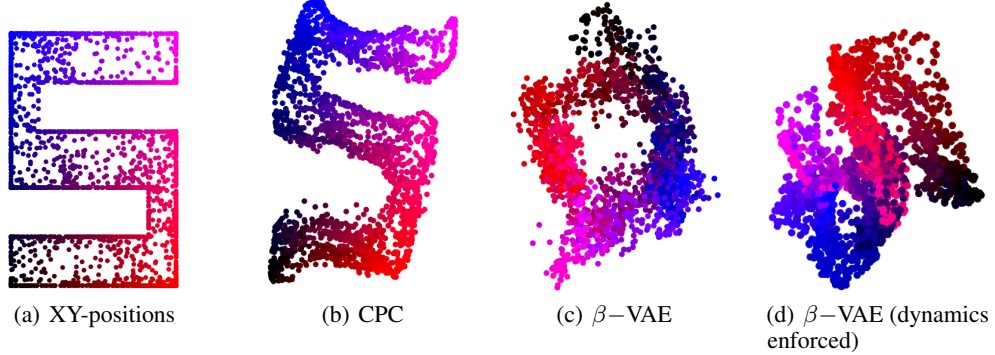

(a) XY-positions      (b) CPC      (c) $\beta-$VAE      (d) $\beta-$VAE (dynamics enforced)

**Figure 11:** Isomap embeddings of learned latent spaces using CPC and $\beta-$VAE for the *BlockS* task.

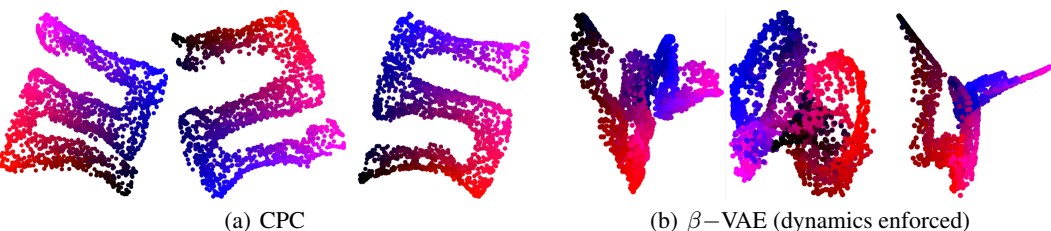

(a) CPC          (b) $\beta-$VAE (dynamics enforced)

**Figure 12:** Isomap embeddings (3 seeds each) of latent spaces using CPC and $\beta-$VAE (with enforced dynamics) for the *BlockS* task.

### D.2 The importance of rejecting improbable transitions during planning

We determine the importance of rejecting unlikely transitions during tree expansion and rewiring by evaluating a modified version of ELAST without such rejection for the environments *BlockS-hard*,

*BlockAsym*, *Button*, *Cable* and *CableParam*. The numerical results of this evaluation in terms of average success rates are presented in Fig. 13. A significant drop in performance was observed for all tasks, which is particularly noticeable for the *BlockS-hard* and *Cable* problems. These results confirm that the transition-density model $\psi$ is a necessary component and effectively recovers the connectivity of the latent states.

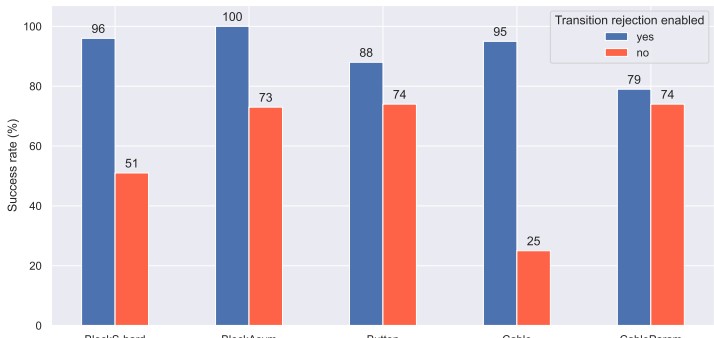

**Figure 13:** Average success rates for *BlockS-hard*, *BlockAsym*, *Button*, *Cable* and *CableParam* tasks for ELAST with and without transition rejection during planning.

### D.3 The effect of the dynamics ensemble

Our method computes latent paths by connecting and rewiring nodes that are obtained through state expansion. Consequently, the quality of the planned solutions depends heavily on the ability of the dynamics model to generate valid states and transitions. In Sec. 4.3, we introduced an ensemble model $H_f$ to further improve the accuracy of the approximated latent dynamics. In order to quantify the benefit of using $H_f$, we evaluated ELAST with a single dynamics model for the environments *BlockS-hard*, *BlockAsym*, *Button*, *Cable* and *CableParam*. The corresponding numerical results are shown in Fig. 14. As expected, the average success rates decreased in all cases except in the *BlockAsym* task.

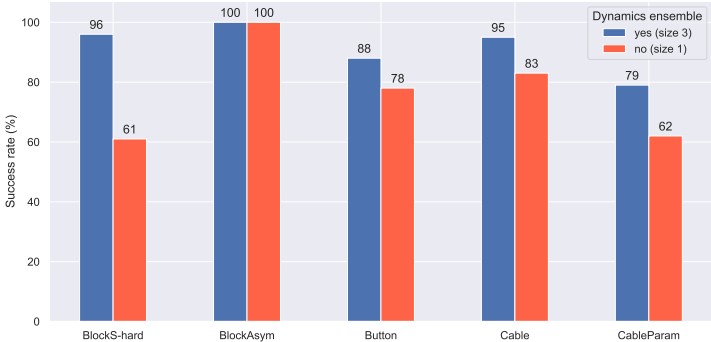

**Figure 14:** Average success rates for *BlockS-hard*, *BlockAsym*, *Button*, *Cable* and *CableParam* tasks for ELAST with and without using the dynamics ensemble during planning.

### D.4 The impact of rewiring the search tree

Our planner reconnects nodes in the search tree in order to improve the overall traveling distances along the tree. To confirm the effectiveness of this rewiring mechanism, we repeated the experimental evaluation with ELAST without rewiring enabled for the environments *BlockS-hard*, *BlockAsym*, *Button*, *Cable* and *CableParam*. Fig. 15 shows that the number of environment interactions that were required to finish the tasks significantly increased without rewiring. Note that this also led to a drop in success rates (see Fig. 16) due to exceeding the maximum allowed number of steps per environment (6).

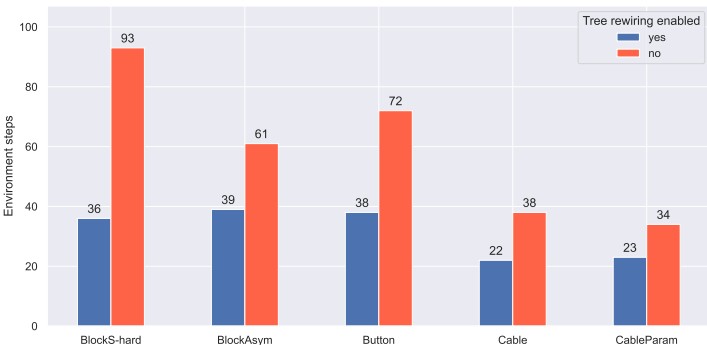

**Figure 15:** Number of environment steps (successful trajectories only) for *BlockS-hard*, *BlockAsym*, *Button*, *Cable* and *CableParam* tasks for ELAST with and without tree rewiring.

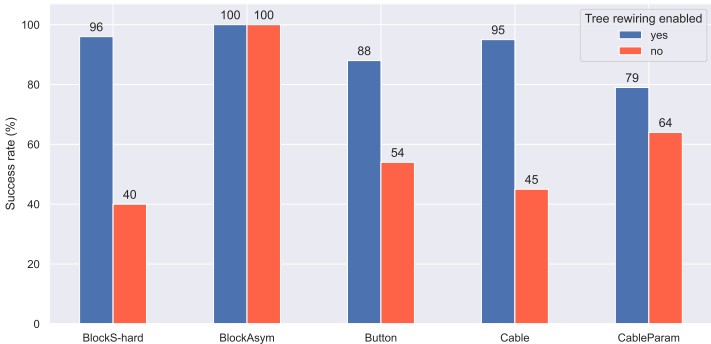

**Figure 16:** Average success rates for *BlockS-hard*, *BlockAsym*, *Button*, *Cable* and *CableParam* tasks for ELAST with and without tree rewiring.

### D.5 How does sampling nodes in sparse regions influence the exploration behavior?

The planning hyperparameters $p_{\text{sparse}}$, $p_{\text{uniform}}$ and $p_{\text{goal}}$ were described in App. A.1.2. To demonstrate that the introduction of the sparse sampling bias $p_{\text{sparse}}$ quickens the exploration of distant areas of the latent space, we computed the success rates of ELAST without this type of sampling enabled. Fig. 17 shows the corresponding numerical results for the environments *BlockS-hard*, *BlockAsym*, *Button*, *Cable* and *CableParam*. As shown, We observed a strong decrease in performance on all tasks in particular for the *BlockS-hard* and *BlockAsym* tasks.

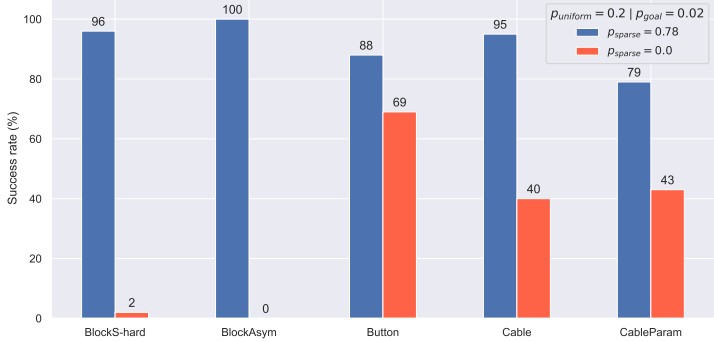

**Figure 17:** Average success rates for *BlockS-hard*, *BlockAsym*, *Button*, *Cable* and *CableParam* tasks with ELAST when sampling nodes in sparse areas is enabled ($p_{\text{sparse}}{=}0.78$) compared to when it is disabled ($p_{\text{sparse}}{=}0.0$).

Fig. 18 visualizes the explored states in the latent space of the *BlockAsym* environment after planning with ELAST for different values of $p_{\text{sparse}}$. Without encouraging sampling in sparse areas ($p_{\text{sparse}}{=}0.0$,

Fig. 18a), the search tree does not explore deep enough to connect to the goal on the other side of the corridor. Yet, for larger values of $p_{sparse}$, the planner achieves better coverage of the latent space resulting in valid paths.

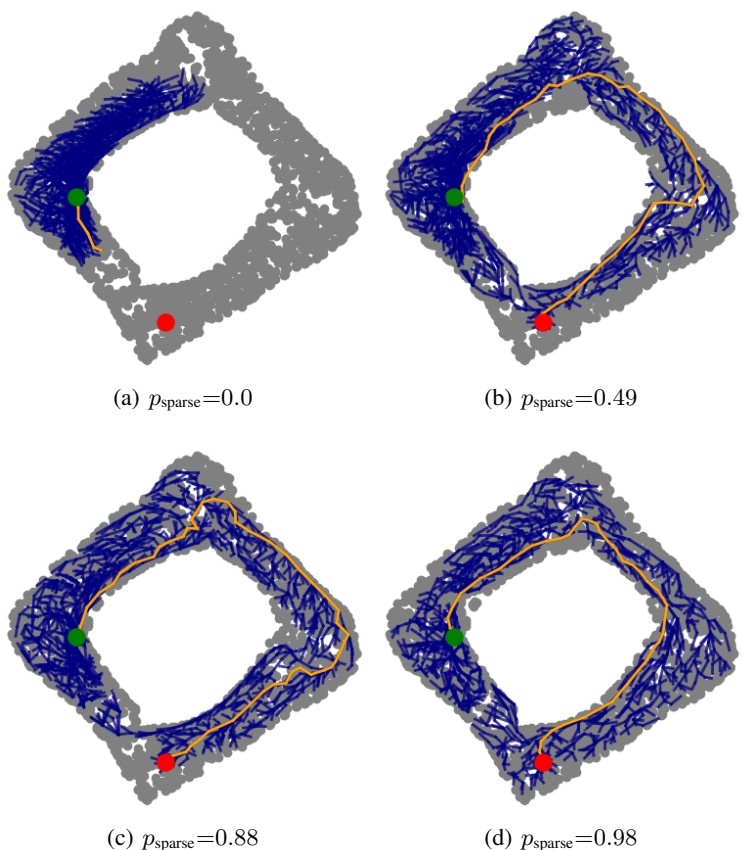

(a) $p_{sparse}=0.0$        (b) $p_{sparse}=0.49$

(c) $p_{sparse}=0.88$        (d) $p_{sparse}=0.98$

**Figure 18:** Impact of sampling bias $p_{sparse}$ for the *BlockAsym* environment. Figures show final paths (orange) and search tree (blue) for $n_{iter} = 5000$, $p_{goal} = 0.02$ and $p_{uniform} = 1.0-p_{goal}-p_{sparse}$ (visualized using Isomap).

### D.6 Computation time of planning module

To assess the practicality of our planner, we measure the average computation time (wall clock) for a new planning query. Note that a comparison with other baselines should be done with caution due to differences in design and hyperparameters. For example, ELAST provides global solution paths that allow for lower replanning frequency ($n_{replan}=50$) compared to trajectory optimization with, e.g., CEM ($n_{replan}=15$). We test both algorithms on a system with an Intel Core i7-8750H CPU and a GeForce GTX 1050 Ti GPU. Due to the iterative processing of ELAST in its current implementation, we evaluate it with the CPU only. However, it should be noted that it could benefit significantly from GPU computation by parallelizing tree expansion at several different nodes simultaneously. We report the runtimes of ELAST in its current implementation and leave the optimization of the code to future investigations.

Fig. 19 shows the average runtimes of ELAST and CEM (GPU implementation) for the *BlockS* and *Drawer* environments. To provide a meaningful comparison, we evaluated both methods with different planning hyperparameters and interpret the relationship between runtime and success rate. As shown, ELAST achieves better success rates even for versions of CEM with higher average computation time per query. Interestingly, we see that the performance of CEM decreases for a large number of samples, which we attribute to the fact that the risk of the planner exploiting invalid transitions increases.

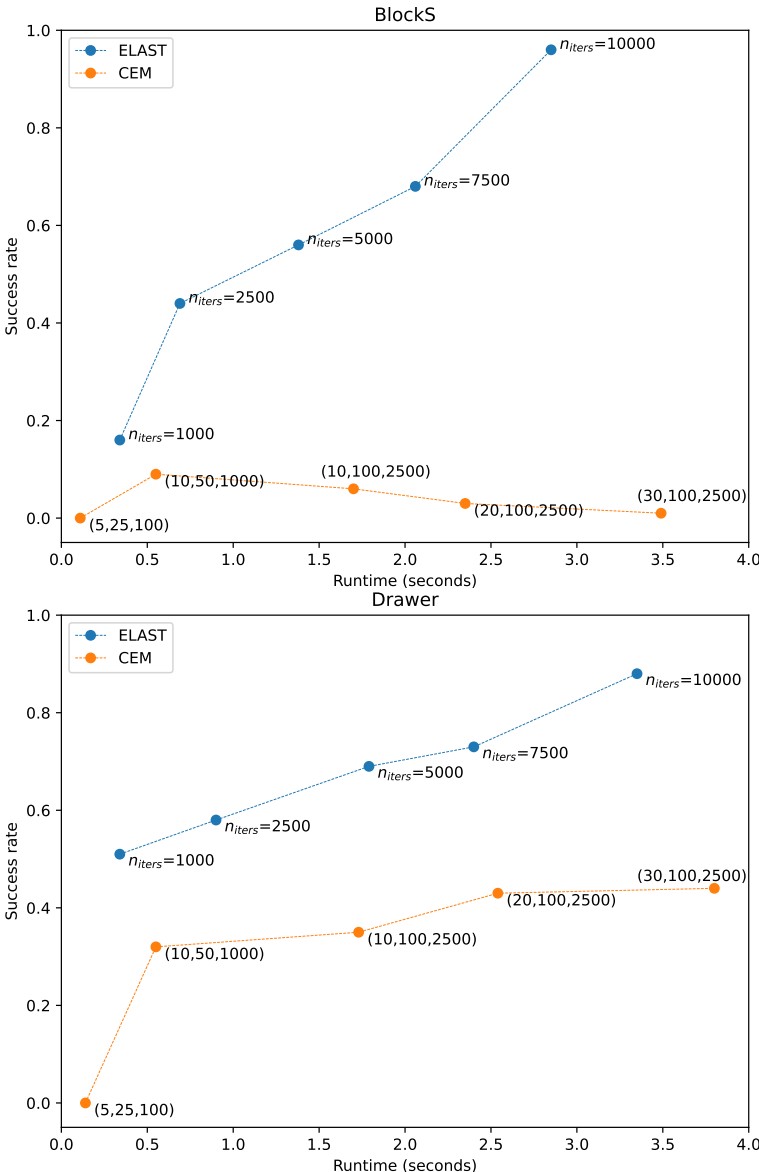

**Figure 19:** Runtime and success rate of ELAST and CEM for the *BlockS* and *Drawer* tasks and for different planning hyperparamters $n_{iter}$, respectively (I,H,K). Results present average over three seeds and 100 test cases-

## D.7 Towards more complex environment parameterizations

Many interesting control settings in robotics are characterized by a large number of environment variations. In our experiments, we use learned encodings of image observations to generate contextual information about the location of obstacles. Using raw image observations instead of exact poses and geometries is advantageous because we bypass the problem of state estimation. In section 5.2, we evaluated our method on environments with a relatively low-dimensional underlying parameterization characterized by the location of few obstacles. To test whether our method can plan robust latent paths for more complex environmental variations, we implemented a version of *BlockParam* for obstacle maps containing a larger number of obstacles ($k=8$), each with varying position, orientation, and shape of rectangular obstacles. Fig. 20 shows several instances explored for this environment. We generated a dataset of 1000 contexts with 20 trajectories of length 25 per context using a random policy. We trained ELAST on this dataset and evaluated it on 100 test cases where the block start and target positions were randomly selected, but using only scenarios with a minimum initial distance of 3 (boxed workspace of size 4x4).

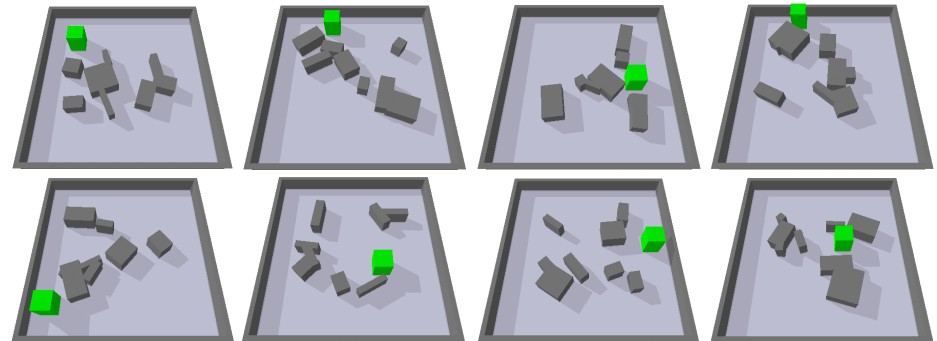

**Figure 20:** Examples of the modified *BlockParam* environment with more complex obstacle maps.

We use a standard autoencoder (convolutional) neural network with a latent dimension of 16 to compress the raw image context observations, and use the resulting encodings as context vectors within our planner. Overall, we observed an average success rate of $85 \pm 5\%$ (3 seeds), demonstrating that ELAST is capable of handling more complex environment parameterizations such as random obstacle maps. To visually support our results, we present the Isomap embeddings and examples of planned latent paths in Fig. 21.

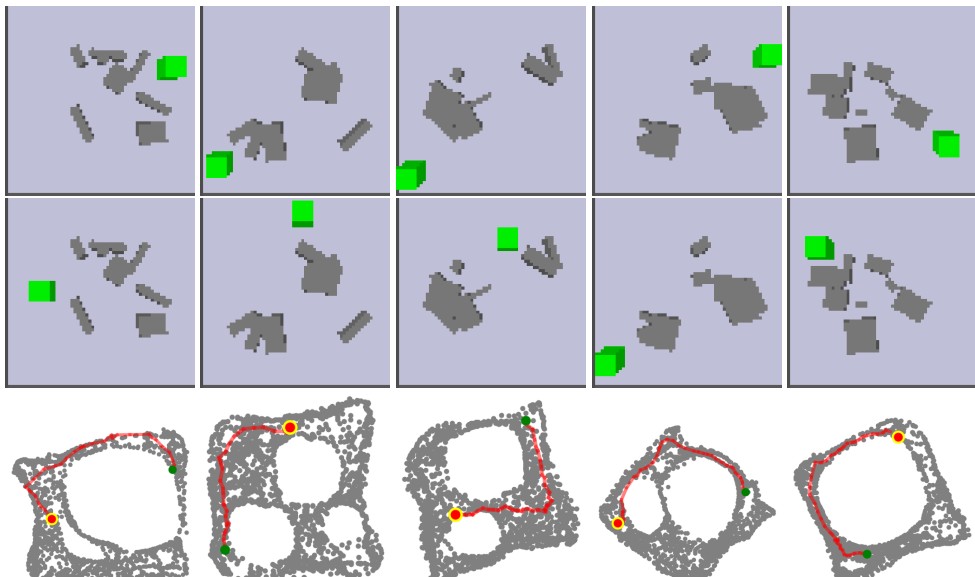

**Figure 21:** Examples of Isomap embeddings and planned paths for the *BlockParam* environment with more complex obstacle maps. Top row: start image, center row=goal image, bottom row: latent path.

**D.8 Comparison of latent exploration between ELAST and CEM**

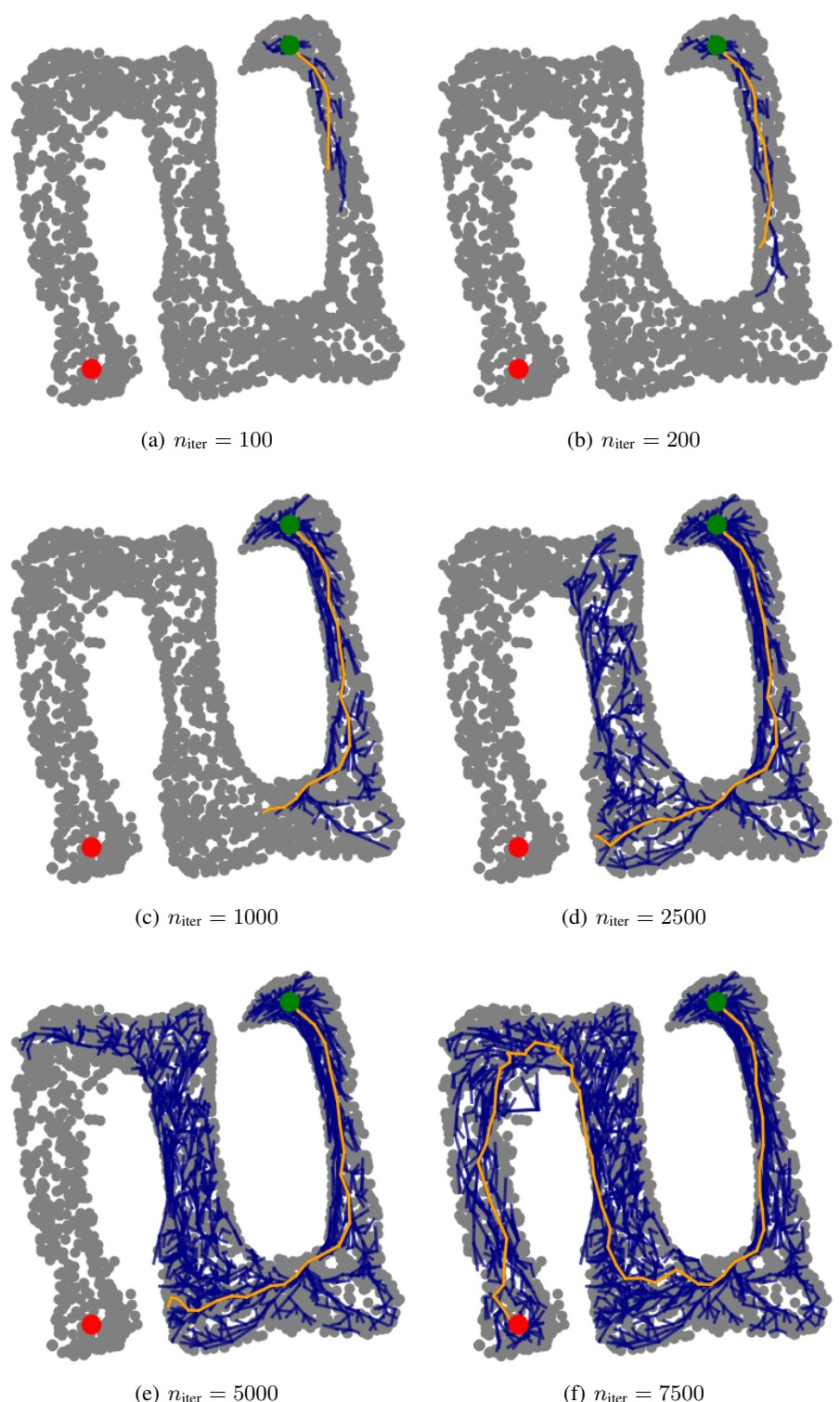

(a) $n_{\text{iter}} = 100$

(b) $n_{\text{iter}} = 200$

(c) $n_{\text{iter}} = 1000$

(d) $n_{\text{iter}} = 2500$

(e) $n_{\text{iter}} = 5000$

(f) $n_{\text{iter}} = 7500$

**Figure 22:** Illustration of explored areas of the latent space (projected into 2D Isomap embeddings) with ELAST for the *BlockS* task. Search tree (blue) and path (orange) for different $n_{\text{iter}}$.

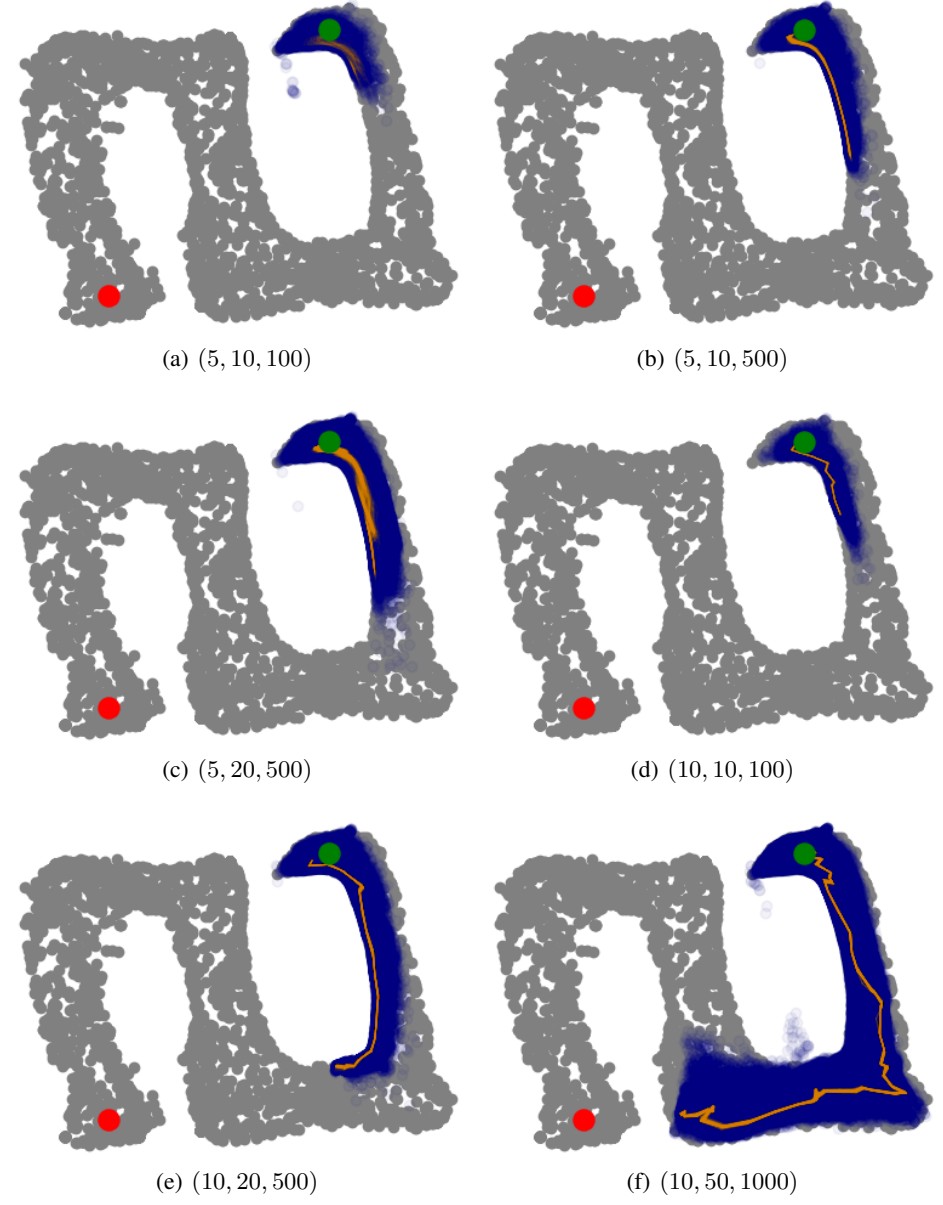

**Figure 23:** Illustration of explored areas of the latent space (projected into 2D Isomap embeddings) with CEM for the *BlockS* task. Visited area (blue) and current population (orange) for different $(I, H, K)$ with iterations $I$, horizon $H$ and $K$ trajectories per iteration.

## D.9 Illustrations of successful latent paths (projecting into 2D with Isomap)

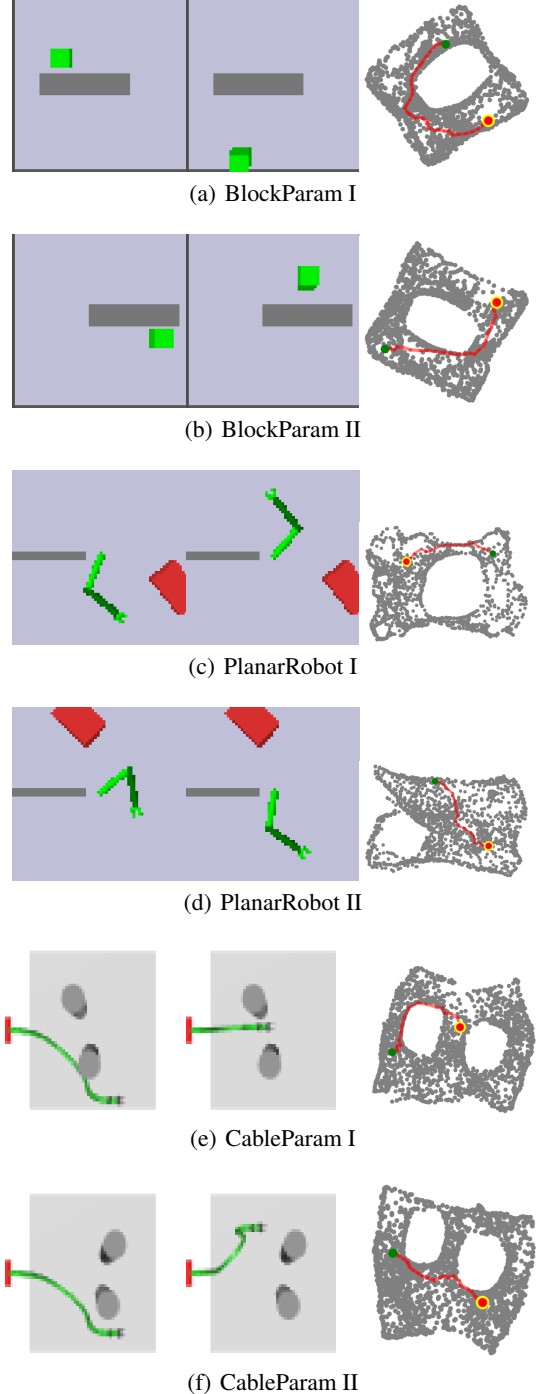

(a) BlockParam I

(b) BlockParam II

(c) PlanarRobot I

(d) PlanarRobot II

(e) CableParam I

(f) CableParam II

**Figure 24:** Successful paths planned with ELAST (parameterized environments). Left to right: start, goal, solution path in Isomap embedding.

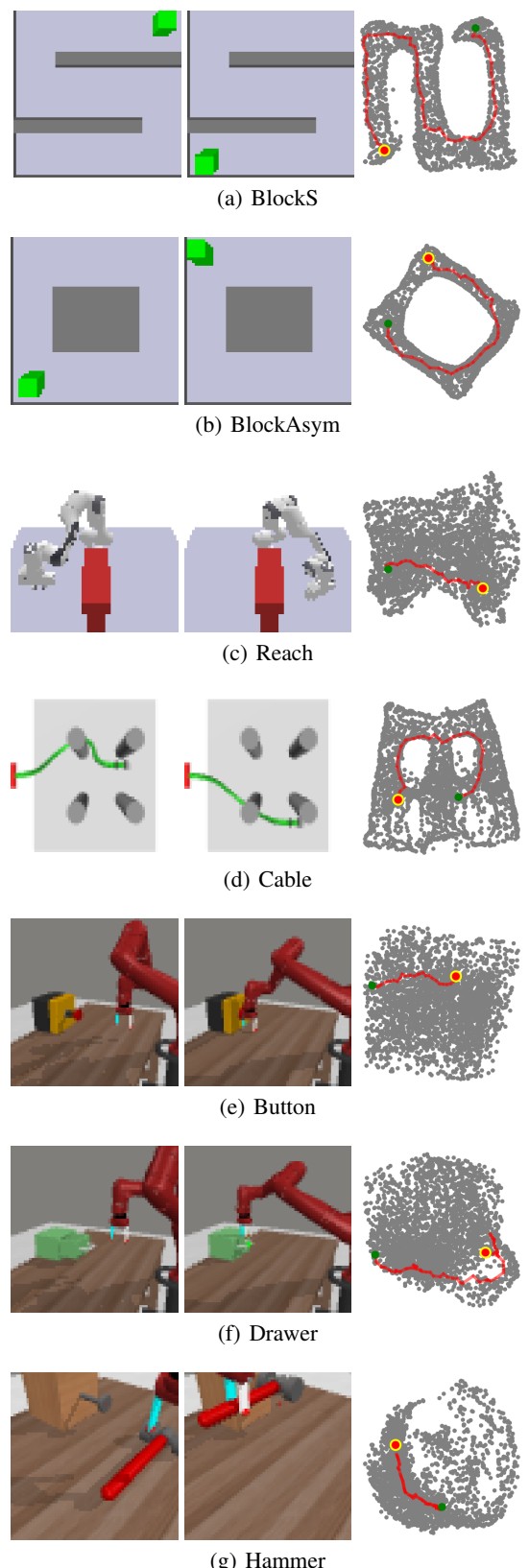

(a) BlockS

(b) BlockAsym

(c) Reach

(d) Cable

(e) Button

(f) Drawer

(g) Hammer

**Figure 25:** Successful paths (unparameterized environments) planned with ELAST. Left to right: start, goal, solution path in Isomap embedding.

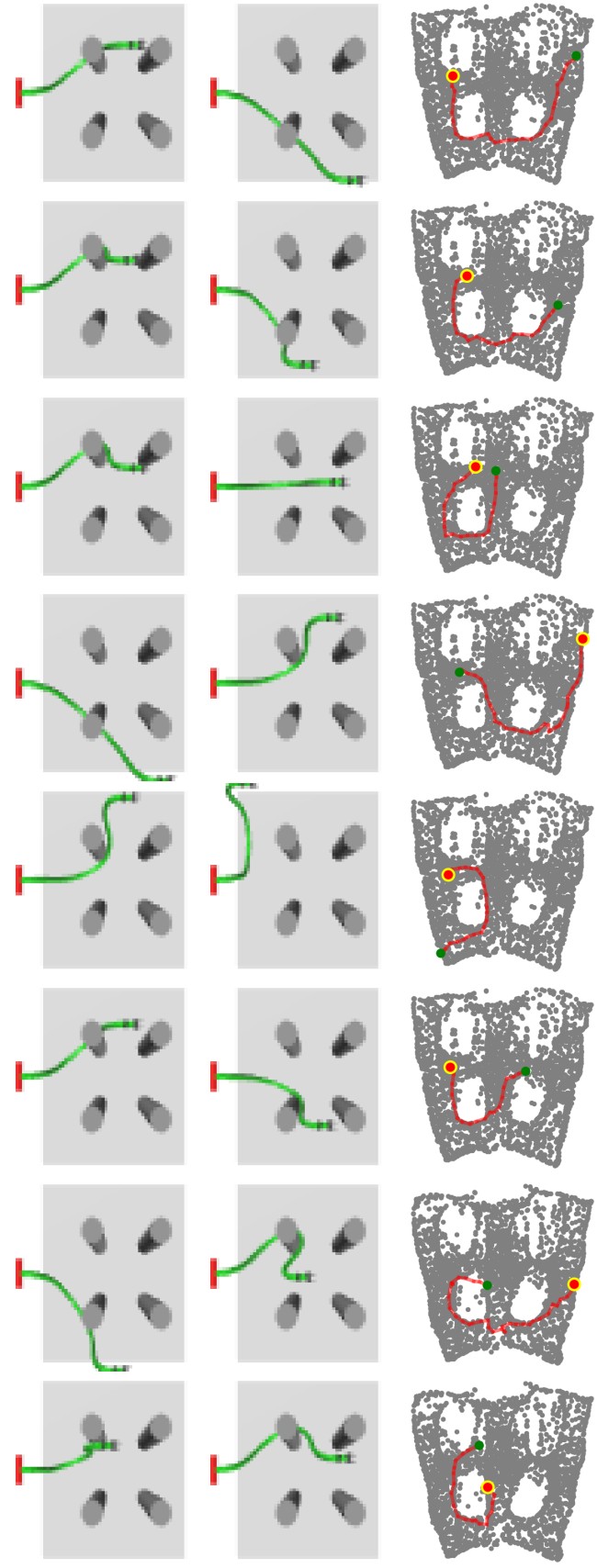

**Figure 26:** Examples of planned latent paths for the *Cable* environment using ELAST. Left to right: start image, goal image, path projected into 2D Isomap embedding.

## D.10 Illustrations of failure cases

We observed that unsuccessful trajectories with ELAST are often caused by improper solution paths containing either invalid transitions outside the latent manifold or large jumps between subsequent nodes. Such invalid transitions can be caused by remaining approximation errors of the latent dynamics and transition density models. Fig. 27 illustrates some examples of unsuccessful latent paths planned with ELAST and projected into the Isomap embedding.

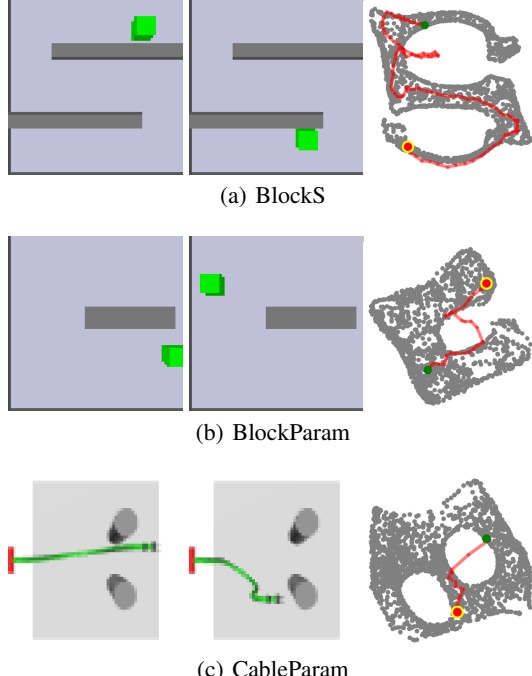

(a) BlockS

(b) BlockParam

(c) CableParam

**Figure 27:** Unsuccessful paths planned with ELAST. Left to right: start, goal, path in Isomap embedding.

For the *Hammer* environment, we found that unsuccessful trajectories are often caused by the planner's inability to generate a global solution path (see Fig. 28). One explanation for this is that ELAST currently uses random action sampling, while the grasping subtask requires selection from a small set of actions that lead to regions of the search space where the object is grasped. Future work could incorporate action sampling heuristics to direct the exploration towards latent states that are likely to be important for solving the task. In addition, we found that one of the main causes of failure in the *Hammer* environment was the agent's inability to reliably grasp the object (see Fig. 29). In the future, the current 1-step policy could be replaced by a more sophisticated control policy, e.g., trained with RL, to achieve greater robustness to the effects of compound errors and improve the performance on manipulation tasks that require a certain degree of dexterity.

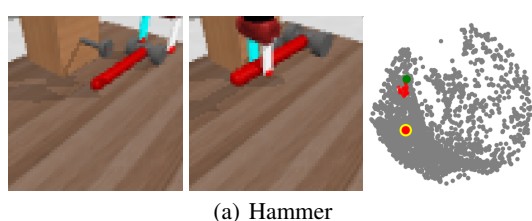

(a) Hammer

**Figure 28:** Unsuccessful path planned with ELAST. Left to right: start, goal, path in Isomap embedding.

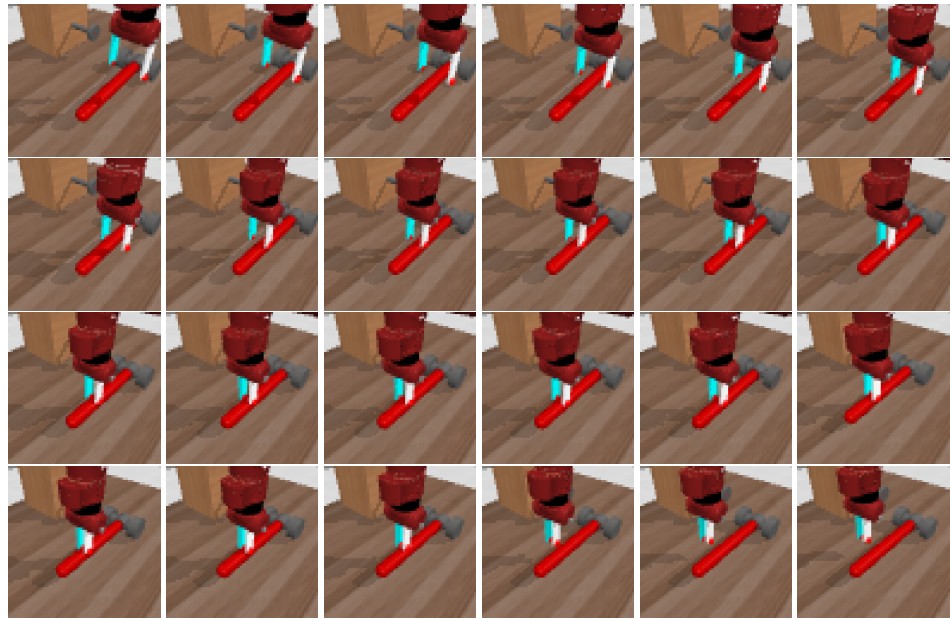

**Figure 29:** Example of unsuccessful grasp in the *Hammer* environment.