# OpenReview forum: "Latent Planning via Expansive Tree Search"
_NeurIPS.cc/2022/Conference — NeurIPS 2022 Accept_

### Official Review · Reviewer_b7bj · 2022-07-10

**Rating:** 6
**Confidence:** 4
**Soundness:** 3 good
**Presentation:** 3 good
**Contribution:** 3 good

**Summary:**

This paper presents a method to perform planning in latent space for robotic applications. The latent space is created for visual observations. The planning procedure follows a methodology similar to sampling-based motion planning in robotics: sampling new configurations (here, in latent space), evaluation of the feasibility of transitioning, re-evaluating the connectivity of this new node to existing nodes. The method is demonstrated in several tasks in simulation, including non-rigid object manipulation, a domain outside of what classical sampling-based planners could solve.


**Questions:**

- Why is the state a sequence of images? What are those sequences for the initial and final states (e.g., s_start and s_goal in Fig.2)?
- What is f in line 197? Please, explain.
- What is the distance metric for the neighbor search in Rewiring? L2 in latent space?
- As I understand it, in line L263 there is a “hidden” implicit assumption: the distance from the tree node in latent space to the goal can be computed in Euclidean space. Or in other words, there is a transformation from latent, to a state where the distance to goal state can be computed by Euclidean distance. Is this true? This seems to strongly restrict the application domains for this method. Also, the effect of this trick is not evaluated in the experiments, but I suspect it provides a large boost. Seems to be however strongly against the idea of planning in latent space? Also, the sentence is weirdly written. The appendix explains another use of Euclidean distance in sampling.
- The experiments are restricted to an MPC setup. Why? As the authors argue, previous approaches could suffer from local minima entrapment. One of the advantages (the main one?) of sampling-based planning techniques such as RRT is that they are able to find a full path to the goal. Why use it in an MPC methodology? Why not evaluate the execution of such a path. Are there issues with it?


**Limitations:**

I indicated some of my doubts about the limitations in the questions. I think some of those could be added to the (currently very brief) discussion of limitations in the paper.

**Strengths And Weaknesses:**

- Significance: the method has the potential to open new avenues of research in planning. It tries to extend a well-studied solution from robotics (sampling-based planning) into domains where they couldn’t be applied, and also reducing the necessary information by integrating it with learning based methods.
- Quality: the paper is well written and easy to read. Minor: starting the paper with “Over the past decade, machine learning has significantly improved the state-of-the-art in image-based robotics and perception” but then citing a single paper feels a bit limited. Consider citing several seminal works, or a survey. Also, maybe there is a better citation than Minsky for the limitations of RL (credit assignment in sparse reward setup). Finally, maybe “unexplored” is better word than “undiscovered” in some places of the text? The planner “explores” these areas to check for feasibility. Also, it should be clear what “undiscovered/unexplored” means in this context. Are these areas where the latents are generated before observing images (extrapolation)?
One notable lack are theoretical warranties. Sampling-based motion planning is a field where these warranties were carefully studied. I wonder about the effect on completeness of sampling using a sampler based on an ensemble of observed dynamics. This seems to steer search towards observed transitions, which indicates that all paths need to be randomly explored at the beginning? Experiments on this would be necessary.
- Originality: the main idea (trajectory planning through sampling in the latent space) has been presented before, as the authors review. However, the presented method is original as it does not require a collision checker and can be used for environments with changes that can be parameterized.
- Clarity: the paper and the concepts are mostly clear. The section about NCE could be improved. For example, the equations are copied from the original NCE paper but the terms are not explained. Some sentences are a bit convoluted, with multiple unclear subordinates (e.g., L151). The use of random trajectories to train is first indirectly introduced in L191; this methodology for training should be explained before, maybe at the beginning of Section 4. In general, the use of demonstrations is not clear. Consider indicating with bold text the best solution in Table 1.

There is a couple of typos, e.g., l117, l207, l220, caption Fig.4,...

---

> ### Author Response · Authors · 2022-08-02
> **Answer to Reviewer b7bj**
>
> Dear reviewer,
>
> Thank you for reading our work and for your detailed comments. Please see the following response to your questions:
>
> >Starting the paper with “Over the past decade, machine learning has significantly improved the state-of-the-art in image-based robotics and perception” but then citing a single paper feels a bit limited.
>
> > Maybe there is a better citation than Minsky for the limitations of RL (credit assignment in sparse reward setup)
>
> Several new references have been added to support the statement of the sentence. We have also added a newer reference for the problem of credit assignment in RL. Please see the revised version of our article.
>
> >It should be clear what “undiscovered/unexplored” means in this context. “Are these areas where the latents are generated before observing images (extrapolation)?
>
> The planner uses a dynamics model to iteratively generate new latent states after observing the initial and goal observations. We use the terms "undiscovered" and "unexplored" regions interchangeably to describe volumes of latent space not yet reached by the successively growing search tree during planning. Given the additional space in the final version, we would like to clarify our choice of words in the main part of the article.
>
> >lack of theoretical warranties
>
> Investigating theoretical results for completeness represents a very interesting direction that we have not explored so far. Unlike classical motion planning, our dynamic models and states are approximated from data, which makes it difficult to derive guarantees. However, if we assume that the learned models are accurate, our method becomes similar to (kinodynamic) EST/RRT, whose completeness properties have been studied in the past (e.g., [1]).
>
> >Why is the state a sequence of images? What are those sequences for the initial and final states (e.g., s_start and s_goal in Fig.2)?
>
> The states consist of three consecutive frames to capture dynamic information such as velocities. We examine goal reaching tasks, where the goal is essentially described by a state of the MDP. Thus, the initial and target states correspond to image sequences, which the encoder then maps to individual latent encodings. Note that in the experimental evaluation, we choose output states and targets with near-zero velocity (nearly identical images in the sequence).
>
> > What is f in line 197? Please, explain.
>
> $f$ is a model of a density ratio $\frac{p(s_{t+k}|s_t)}{p(s_{t+k})}$ which is optimized to preserve the mutual information between $s_{t+k}$ and $s_k$ (up to multiplicative constant). In essence, this model describes a positive score (also see [2]) which resembles the similarity between states. In our case, we use the exponential of the negative squared Euclidean distance. Unlike [2], we choose this distance in order to be more in line with the way we perform nearest neighbor search (Euclidean distance) in the latent space.
>
> > What is the distance metric for the neighbor search in Rewiring? L2 in latent space?
>
> Yes, we use the L2 metric in latent space to determine a preliminary set of neighboring nodes.
>
> > As I understand it, in line L263 there is a “hidden” implicit assumption: the distance from the tree node in latent space to the goal can be computed in Euclidean space. Or in other words, there is a transformation from latent, to a state where the distance to goal state can be computed by Euclidean distance. Is this true?
>
> We would like to emphasize that ELAST **does not assume access to the ground truth temporal or spatial distances** between latent states and the goal. The Euclidean distance is only used as a heuristic (with $p_\textrm{goal}$) to pick a the next node for expansion which is not necessarily the nearest to the goal in terms of the remaining number of steps. This heuristic is merely useful to give the exploration a meaningful direction. Note that similar strategies are common in classical sampling-based motion planners such as RRT. We have revised the relevant passages in the text.
>
> > The experiments are restricted to an MPC setup. Why?
>
> We do not have an explicit representation of the environment and thus cannot directly compute the quality of solutions without simulation. Instead, we evaluate the utility of the latent paths for downstream control tasks. Running without online replanning would lead to open-loop control. In this case, compounding errors (due to model inaccuracies, i.e., dynamics+states+policy) lead to a deviation from the planned path. A simple way to improve robustness is to embed ELAST in an MPC loop, i.e., re-plan after a few steps, thus closing the feedback loop. Note that this is a common approach used by relevant baselines (e.g. HTM) and also facilitates comparison with e.g. CEM.
>
> [1] Probabilistic completeness of RRT for geometric and kinodynamic planning with forward propagation, Kleinbort et al., RA-L 2019
>
> [2] Van den Oord et al.. Representation learning with contrastive predictive coding. CoRR, 2018

---

> > ### Comment · Reviewer_b7bj · 2022-08-08
> > **Reply to comments**
> >
> > Thanks for your clarifications.
> >
> > I'm still a bit confused by the "hidden" assumption. Yes, ELAST does not assume distances, but your full system (and the one you report best results for) uses it for the sampling procedure. Without the goal directed sampling, results change quite some. And yes, classical planners use that heuristic because they plan (sample, check) in Cartesian space, so nothing weird here, but your planner does it in latent space.

---

> > > ### Author Response · Authors · 2022-08-08
> > > **Clarification regarding  "goal" sampling**
> > >
> > > Dear Reviewer,
> > >
> > > Thank you for your comment. We would like to clarify and emphasize that **no hidden assumptions** are used, neither in our planner nor in its evaluation (full system). For all experiments, we report the results **without** access to the underlying distances. We hope that the following explanation can help clear up any possible misunderstandings.
> > >
> > > Given our initial observation and target observation, our encoder $\phi$ generates the corresponding initial latent encoding $z_\textrm{start}$ and target latent encoding $z_\textrm{goal}$. Our planning module then searches for a connecting path between $z_\textrm{start}$ and $z_\textrm{goal}$ by expanding and rewiring a search tree within the latent space $\mathcal{Z}$. At each expansion step, a node $z_\textrm{expand}$ in the existing tree is selected and a node is created from it using a random action. To explore efficiently, we use two sampling heuristics to choose $z_\textrm{expand}$: (a) one that increases the chance of selecting nodes in sparsely represented areas, and (b) one that simply selects the node in the current tree that is closest to the target state $z_\textrm{goal}$ with respect to the Euclidean distance in $\mathcal{Z}$, i.e., the **Euclidean distance between two latent vectors**. Note also that the latter does not require that the Euclidean distance be equal or close to the true underlying distance to the target $z_\textrm{goal}$. Nevertheless, it has been found useful to add a small fraction of 2% of this strategy to slowly steer the search towards $z_\textrm{goal}$. Note that increasing this fraction clearly has detrimental effects on the planning results, as the planner would act too greedily with respect to this heuristic and often get stuck in local minima.
> > >
> > > Apart from these two heuristics, we do **not** use any other form of sampling bias. All of our reported results are based on the strategy described above, which does not require access to the underlying distances.
> > >
> > > We are happy to answer any further questions and provide additional explanations to help address any remaining concerns.

---

### Official Review · Reviewer_ASfb · 2022-07-11

**Rating:** 6
**Confidence:** 4
**Soundness:** 2 fair
**Presentation:** 2 fair
**Contribution:** 3 good

**Summary:**

The authors propose ELAST, a method for model-based long-horizon planning in a low-dimensional latent space of partially-observable systems with high-dimensional observations $s_t$ (e.g. images).
The method can be separated in three conceptual stages.
- First, latent encodings $z_t$ are obtained for each observation $s_t$ (see below).
- Second, a transition model is fit to time-consecutive encodings to capture the system dynamics $p(z_{t+1} | z_t)$ (disregarding context $c$ for brevity here).
- Finally, the transition model is used for latent space planning towards a latent goal.
This is done by expanding a search tree and then using a heuristic to cull expanded nodes based on the transition model's density, such that the tree remains compliant with the learned dynamics (i.e. sampling-based motion planning, inspired by Rapidly-exploring Random Trees (RRT) and Expansive Space Trees (EST)).
The low-dimensional latent space lowers the search cost, which allows the authors to use long planning horizons.
The search produces waypoints for an MPC controller, which uses a local parametric policy to bridge them.

The method is validated over 10 different simulated environments with decent complexity and reasonable success rates.

In terms of model learning, the latent encodings are a byproduct of applying *Contrastive Predictive Coding* (CPC) [1], to estimate the density ratio $p(s_{t+k} | s_t) / p(s_{t + k})$ which implicitly captures the dependence between consecutive observations $s_t$.
The objective of CPC is called InfoNCE and is heavily inspired by *noise contrastive estimation* (NCE) [2] and avoids generative modelling of the observations $s_t$ alltogether.
The encodings alone are not enough for planning, as a latent transition model is also necessary.
The authors estimate the transition's conditional using NCE as well in a second step, based on the latent encodings alone.
There is some redundancy in modelling the transition (see my comments below).

**References**

[1] Oord, A.V.D., Li, Y. and Vinyals, O., 2018. Representation learning with contrastive predictive coding. arXiv preprint arXiv:1807.03748.

[2] Gutmann, M. and Hyvärinen, A., 2010, March. Noise-contrastive estimation: A new estimation principle for unnormalized statistical models. In Proceedings of the thirteenth international conference on artificial intelligence and statistics (pp. 297-304). JMLR Workshop and Conference Proceedings.


**Questions:**

- Using RRT / EST trees for expansion leads to heuristics, e.g. the rewiring or the heuristic bias towards unexplored regions. Have you considered search that is better aligned with dynamic programming?

Also see my remarks above.


**Limitations:**

I don't see any major negative social impact. Some of the method's limitations are acknowledged in a dedicated section.

**Strengths And Weaknesses:**

**Strengths**
- The method seems widely applicable, as little domain knowledge is employed.
- Planning appears to be efficient, the authors consider long search horizons and up to 7500 tree expansion steps during planning.
- Using search for control, be it with tree heuristics like here or through optimal dynamic programming, is generally better than local optimization methods that are prone to getting stuck in local minima.
- The formulation appears novel to me.

**Weaknesses**
- The paper sets out to address planning for partially-observable systems, specifically where the state-space does not match the observations, which for me implies a Partially-Observable Markov Decision Process (POMDP).
However, the methods are geared towards solving an MDP, as lookahead control (MPC) is used and generative observation modelling is omitted alltogether.
I found no clarification of this aspect in the main text, and I believe this is a very important point in terms of positioning the method and making its limitations clear.
For example, an optimal POMDP controller would need to plan in *belief space*, and account for the effect of future observations on the state estimate [1].
State estimates should also generally be uncertain, to account for observation noise and modelling errors, and from what I can tell the obtained latent encodings are a deterministic mapping of the observations.
Could you please comment on how the method fits / differs from the POMDP paradigm? I believe this should be discussed in the manuscript.
- I can see how observation generation can burden model training, but why is that an issue during deployment for control (sec. 1, paragraph 2)? Assuming that we disregard the POMDP aspect (which is currently the case), control would only require the dynamics and rewards / the goal state.
- Contrastive estimation to avoid observation generation has been considered in Dreamer [2] before, where through a single objective dynamics are learned in parallel with shaping the latent space (I'm referring to the mutual-information NCE variant, not the ELBO). What is the justification of learning the transition in two stages (first the encodings through CPC, then the dynamics through NCE)?
- $h_f$ (the forward dynamics model) and $\psi$ (the transition density) seem redundant, and this redundancy raises questions. E.g. why is the transition density learned in a second stage (see comment about Dreamer above) and why isn't the forward dynamics model trained to capture the transition density already?
- It is argued that NCE is better for transition estimation because the noise model provides outliers (end of sec. 4.2). But the noise model should be as close to the data distribution as possible for unbiased NCE gradients--can you elaborate?
- Thresholding based on the transition density when rejecting samples from the search tree does not properly capture the probabilistic mass of the transition (sec. 4.3). Have you considered rejection sampling / importance sampling?
- The experimental section ablates the planner and shows it is a core requirement for success, but ablations of model learning (encoding, transition) would also be helpful to assess the method. E.g. what would happen if the planning scheme is used with dynamics learned through generative modelling, e.g. PlaNet or Dreamer? Currently one cannot tell if the contrastive modelling is a major factor in terms of success, or generative modelling could work just as well.

**Further Remarks**
- I believe the choice of the similarity used in CPC should be motivated further, with more details about why it enforces proximity preservation. This is not immediately clear from the ratio estimation of $p(s_{t+k} | s_t) / p(s_{t+k})$ alone.
- I did not find details about the training of the ensemble $H_f$, this ties into my comment about the redundant transition modelling.
- Sequences of observations are generally not guaranteed to be Markovian (sec. 4, first sentence).

Despite the above weaknesses, the method appears pragmatic and seems to work on the considered tasks.
I find that search in a low-dimensional latent state-space is generally a good idea, albeit driven by heuristics.
This makes me lean towards a slightly positive assessment.
However, I have reservations about the methodology and the authors' input would be appreciated.

**References**

[1] Bertsekas, D., 2012. Dynamic programming and optimal control: Volume I (Vol. 1). Athena scientific.

[2] Hafner, D., Lillicrap, T., Ba, J. and Norouzi, M., 2019. Dream to control: Learning behaviors by latent imagination. arXiv preprint arXiv:1912.01603.

---

> ### Author Response · Authors · 2022-08-02
> **Answer to Reviewer ASfb**
>
> Thank you for the very detailed feedback. Please see the following response.
>
> > ..for me implies a Partially-Observable Markov Decision Process (POMDP). However, the methods are geared towards solving an MDP, as lookahead control (MPC)..
>
> We consider states as sequences of image observations and do not assume perceptual uncertainty of the encoder. We also assume that the visual data contains enough information to describe the full and next state of the environment. This leads us to an MDP formulation. Nevertheless, we recognize that extending ELAST towards generative observation modeling and planning in belief spaces would be a logical next step for future work. In particular, this allows us to address partially observable environments and model perceptual uncertainty. We would like to give more details about the modeling assumptions in Sec. 4+6 as more space is available in the final version.
>
> > I can see how observation generation can burden model training, but why is that an issue during deployment for control (..)? Assuming that we disregard the POMDP aspect ..), control would only require the dynamics and rewards / the goal state.
>
> Algorithms for planning in image spaces often rely on video generative models, which consist of a large number of model parameters and are therefore slower in synthesizing new states. Low-dimensional latent dynamic models, on the other hand, are usually more lightweight, allowing us to generate and evaluate a large number of states in a short time.
>
> > What is the justification of learning the transition in two stages (first the encodings through CPC, then the dynamics through NCE)?
>
> In the first step, the encoder $\phi$ is trained together with a single dynamics model $h_f$. We have found that the performance of the planner can be improved if a dynamics ensemble is used that is trained on the already learned embedding (see ablation in App. D3). Essentially, the ensemble improves the quality of the model, resulting in fewer false transitions. Note that training $\phi$ and $h_f$ together gives a moving target distribution for dynamics learning. The use of two stages, i.e., fixing the encodings, facilitates optimization, which improves the accuracy of the learned dynamics model.
>
> >$h_f$ and $\psi$ seem redundant, and this redundancy raises questions. E.g. why is the transition density learned in a second stage (...) and why isn't the forward dynamics model trained to capture the transition density already?
>
> $h_f$ is used during the node expansion to generate new states from current states and actions. $\psi$, on the other hand, solves the inverse problem of estimating whether a transition between two states (without actions) is likely given the data. $\psi$ is used to (a) filter out unlikely transitions during expansion and (b) check whether rewiring of nearby states is possible. To combine both in one would require a model that can be easily used in both ways. An example would be normalizing flows, which allow data generation and density estimation. However, our approach relies more on standard network architectures that allow for fast inference and were shown to work well in our experiments. Alternatively, one could use only an energy-based model such as $\psi$ and employ MCMC sampling techniques to generate new states. A major drawback here would most likely be slow inference, which would lead to a computational bottleneck in the planner.
>
> >..NCE is better for transition estimation because the noise model provides outliers (..). But the noise model should be as close to the data distribution as possible for unbiased NCE gradients
>
> NCE works best when the noise distribution closely surrounds the data. We place Gaussian noise at the starting nodes of the transitions, which produces a sufficiently narrow noise distribution. Sampling from this (untruncated) Gaussian distribution also produces points that lie outside the latent manifold (see Fig. 3 Gaussian noise "spilling" beyond the data support). This means that our model is trained for outliers, which is crucial for planning to correctly identify and reject low-density transitions that lead outside the latent manifold.
>
> >Thresholding based on the transition density when rejecting samples from the search tree does not properly capture the probabilistic mass of the transition (..). Have you considered rejection sampling / importance sampling?
>
> Thank you for your suggestion. Note that our thresholding is applied to the conditional transition densities given the start node of a transition. We have so far not considered other rejection strategies but found that simple thresholding works well in our experiments.
>
> >The experimental section ablates the planner and shows it is a core requirement for success, but ablations of model learning (encoding, transition) would also be helpful
>
> We revised App. D1, the impact of the embedding. It tests the impact of an embedding learnded with a VAE with and without enforced dynamics.

---

> > ### Comment · Reviewer_ASfb · 2022-08-07
> > **Thanks for your response**
> >
> > Thank you for taking the time to work through my comments.
> >
> > I believe analyzing the impact of the embedding is a valuable addition (app. D1). Considering a generative state-space model instead of a VAE would have been ideal, but I understand that this would be too much given the time limitations that come with the review process.
> >
> > Thank you for clarifying your motivation behind modelling $h_f$ and $\psi$ separately. A single dynamics conditional would have been more elegant, but I understand the pragmatism. It might be worth pointing this out explicitly in the final version.
> >
> > In terms of generative modelling: planning in image spaces would indeed scale poorly, what I meant is that there are many recent generative state-space models (sometimes referred to as world models) that feature a low-dim latent state space as well. A very recent example is [1]. Planning in such models comes down to using a dynamics conditional that is not too different from the one discussed in this paper and therefore should not burden the search (i.e. observations in such models are only needed for state estimation, not for evaluating the latent space dynamics, disregarding a POMDP formulation). Hence my interest in the impact of the embedding, as it's related.
> >
> > [1] Wu, P., Escontrela, A., Hafner, D., Goldberg, K. and Abbeel, P., 2022. DayDreamer: World Models for Physical Robot Learning. arXiv preprint arXiv:2206.14176.

---

> > > ### Author Response · Authors · 2022-08-08
> > > **Re: Thanks for your response**
> > >
> > > Thank you very much for your comment! We agree that generative modeling of latent dynamics would be computationally feasible. This could also be easily incorporated into our method, e.g., by using standard conditional generative models such as cVAEs [1]. For simplicity and to keep the complexity at a moderate level, this was not implemented. As indicated earlier, however, we are curious to follow the suggestion and integrate generative state and dynamics modeling in the future (similar to [2]). In this regard, it would also be interesting to study how representation learning via contrastive predictive coding [3] is extended towards stochastic dynamics.
> > >
> > > [1] K. Sohn, H. Lee, and X. Yan, Learning structured output representation using deep conditional generative models, NeurIPS 2015
> > >
> > > [2] Danijar Hafner, Timothy Lillicrap, Ian Fischer, Ruben Villegas, David Ha, Honglak Lee, and
> > > James Davidson. Learning latent dynamics for planning from pixels. ICML 2019.
> > >
> > > [3] Aäron van den Oord, Yazhe Li, and Oriol Vinyals. Representation learning with contrastive
> > > predictive coding, CoRR 2018.

---

### Official Review · Reviewer_v4DQ · 2022-07-12

**Rating:** 6
**Confidence:** 3
**Soundness:** 2 fair
**Presentation:** 3 good
**Contribution:** 3 good

**Summary:**

This paper proposes a new algorithm for planning and control for long-horizon goal-reaching tasks in high-dimensional spaces (e.g., control from vision). To do so, the proposed approach performs sampling-based planning in a lower-dimensional latent space. This latent space and its associated transition model are learned using contrastive learning. With these two building blocks, the proposed approach recursively plans a sequence of latent states that are tracked using a learned control policy. The approach outperforms existing baselines in a number of simulation experiments.

**Questions:**

- Computation times: how long does it take for the approach to compute a plan? How does it compare with baselines? What is the computational bottleneck in the proposed approach? How does it depend on the size of the learned latent space representation?

- Comparisons and feedback control: the role of feedback (from replanning with MPC and tracking the plans with the local policy $\pi$) should be clarified in the comparisons. The following important details are not described in the body of the paper or in the appendix, which hinders reproducibility and interpretability of the results:
--- Is PlaNet [12] implemented with online replanning and/or local feedback or not? Is a new trajectory computed at each iteration (as in the original paper) or every 3-25 time steps as in the CPC baselines?
--- Is HTM [29] implemented with the same local policy $\pi$ as ELAST? Is replanning done as in ELAST for a fair comparison, or not?

- The paper mentions the difficulty of planning through narrow passages in the Hammer environment. Is the corresponding region in the learned latent space also narrow or not necessarily? Indeed, a good learned latent representation for sampling-based planning would assign larger volume to feasible "narrow" passages. Does the proposed approach yield this property or not necessarily? Can the authors add the learned latent space for failure cases in the Hammer environment in Appendix D.8?

Suggestions:
- The discussion in the paragraph: "Does the rewiring affect the quality of generated paths?" is not surprising and follows from results in the sampling-based planning literature (rewiring is a standard step). I recommend making this paragraph shorter.
- The paper and appendix include typos and should be proof-read (e.g., previosuly, exisitng, distriubtion, This baselines applies, etc.).

**Limitations:**

Yes, the authors state the limitations of the approach (but, importantly, the paper misses a discussion about the computation times). A discussion of failure cases (e.g., only 22% success rate on the Hammer environment) is included.

If the authors provide a convincing discussion about the computation times for their proposed approach, I will be happy to raise my rating.

**Strengths And Weaknesses:**

This paper makes an interesting contribution for long-horizon planning in high-dimensional spaces, as it combines (long-horizon) sampling-based planning techniques from the robotics community with learned representations from the machine learning community to enable planning in a lower-dimensional latent space. The approach is original and clearly presented, and, in general, the paper is well-written.

** Strengths **
- The learned latent space and density transition model alleviate the need to learn a collision detector, distance metric, and sampling distribution for sampling-based planning.
- By planning and learning a transition model in a latent space, the approach does not require learning high-dimensional transition models (e.g., video generative models), which can be computationally expensive to train and evaluate.

** Weaknesses **:
- The paper does not report computation times. As such, it is unclear if the approach runs in real-time and can be applied to real-world robotic problems, or if it only works in simulation. In particular, more details should be provided regarding scalability of the method with respect to the dimension of the latent space, computation times for the MPC controller, etc.

---

> ### Author Response · Authors · 2022-08-02
> **Answer to Reviewer v4DQ**
>
> Dear Reviewer,
>
> Thank you for reading our work and providing useful feedback. Please see our response below:
>
> > Computation times: how long does it take for the approach to compute a plan? How does it compare with baselines? What is the computational bottleneck in the proposed approach? How does it depend on the size of the learned latent space representation?
>
> We have a new section on computation times in App. D6. Direct comparison of the required computations with the baselines is difficult due to differences in design, control loop frequency, and hyperparameters. We provide an evaluation that considers runtimes and resulting success rates for different hyperparameter configurations to provide a general intuition. Investigating the planning runtime for different latent sizes is indeed interesting. We would like to provide a corresponding experiment in the final version of our paper.
>
> > Comparisons and feedback control: the role of feedback (from replanning with MPC and tracking the plans with the local policy ) should be clarified in the comparisons. The following important details are not described in the body of the paper or in the appendix, which hinders reproducibility and interpretability of the results
>
> For all planning and trajectory optimization methods in the experiments (including PlaNet and HTM), we use online replanning. We do not use local feedback, which means that between replanning we simply execute the computed actions of the current plan one after the other (this strategy was used across all methods where applicable).
>
> > Is PlaNet [12] implemented with online replanning and/or local feedback or not? Is a new trajectory computed at each iteration (as in the original paper) or every 3-25 time steps as in the CPC baselines? -
>
> Yes, we implemented PlaNet with online replanning (no local feedback). We replan every 3-25 environment steps (3 steps for 'Hammer' task, 25 otherwise).
>
> > Is HTM [29] implemented with the same local policy as ELAST? Is replanning done as in ELAST for a fair comparison, or not?
>
> HTM was implemented using the code and architectures of all models from the paper (App. C6). Therefore, we do not use the same local policy as in ELAST. Also, we used the planning hyperparameters **(replanning every 10 steps)** as suggested by the authors.
>
> > The paper mentions the difficulty of planning through narrow passages in the Hammer environment. Is the corresponding region in the learned latent space also narrow or not necessarily? Indeed, a good learned latent representation for sampling-based planning would assign larger volume to feasible "narrow" passages.
>
> In the context of the Hammer environment, we mean by "narrow passages" parts of the state-action space corresponding to the grasping of the object. Note that only a small set of actions yield robust grasps with do not drop the object and allow exploration towards goal states at which the hammer presses down the nail.  ELAST uses the original action space and samples actions uniformly during planning, therefore a "narrow passage" is presented. Currently, we do not explicitly assign a larger volume to narrow passages in latent space.  However, learning such embeddings to facilitate planning through hard-to-explore but feasible regions would be very interesting to explore in the future. Another idea to mitigate the effects of difficult exploration would be to integrate task-informed sampling heuristics for state and action sampling.
>
> >  Can the authors add the learned latent space for failure cases in the Hammer environment in Appendix D.8?
>
> We have added a visualization of a failed plan in App. D10 and also provide an improved discussion of failure cases in this section.

---

> > ### Comment · Reviewer_v4DQ · 2022-08-09
> > **Thanks for your responses**
> >
> > I would like to thank the authors for their detailed responses, in particular regarding computation times. The responses corroborate my positive assessment of this paper.

---

### Official Review · Reviewer_6tyY · 2022-07-16

**Rating:** 6
**Confidence:** 4
**Soundness:** 3 good
**Presentation:** 3 good
**Contribution:** 3 good

**Summary:**

This work presents ELAST, an approach for long-horizon planning through sampling-based motion planning in learned latent spaces. Given a high dimensional observation (images), the approach first learns a latent representation using self-supervised contrastive learning, a latent dynamics model, and a transition density function. This latent space is then searched via expansive space trees to solve motion planning problems from initial states to goal states. Finally, plans are executed via MPC. ELAST allows efficient searching of spaces and long-horizon planning as shown over a number of environments.


**Questions:**


See above in strengths and weaknesses as well
- What is the computation time?
- I think an ablation to VAE-based latent spaces with enforced dynamics is missing.
- How robust is the latent space? It would be good to see N random training seeds in the Appendix shown with isomaps. Similarly this would be informative for VAE-based latent spaces and VAE-based spaces with enforced dynamics.
- Why does 19c/d form a loop in the isomap?


**Limitations:**

I would appreciate greater discussion of the limitations of ELAST, right now it's focused more on future work and how previous works can be incorporated. A few limitations of at least the current setup I see is it is not clear how robust the approach is to more complex environments (e.g., with more obstacles) and higher dimensional problems (in the true low-level state space or in the controls space). It would also be good to understand how it scales to kinodynamic systems. I feel the computation time may too be a limitation depending on the results.

**Strengths And Weaknesses:**

The paper is generally well written and clear, as are the figures demonstrating the approach. The area of learning latent representations is useful and important. The minimal assumptions and requirements on data for ELAST makes the method generally applicable. Based on the planning results and plots of the latent spaces the latent space learned is generally robust. The MPC replanner seems to be effective for following the trajectories.

In terms of number of environments and comparisons the experiments section is thorough (though see notes below on complexity of the environments). The detailed ablations in the appendix and ablations in the experiments section are well done.

The two main weaknesses I see are (1) the complexity of the environments (both in terms robot dimensionality and environment variation) and (2) the novelty compared to prior approaches for latent tree searches.

(1) Each of the environments has relatively low dimensional true, important state spaces, which is an area where these latent space methods perform well. It would be useful to see how it performs in spaces such as an SE(2 or 3) rigid body in a complex environment. Furthermore it would be useful to see how it performs with more complex dynamics, such as a kinodynamic system which may force higher dimensional learned latent spaces (a double integrator for instance).

Finally, as a benefit over prior methods is that it does not require a collision checker (which allows some degree of generalization to new environments) it is imperative the authors demonstrate the approach on more complex obstacle environments, which can be encoded via the context. I’m not sure from the results shown that it would perform well in such cases. In the current implementation, the conditioning on the actual states of parameterization (such as the position of the moved block) does not seem consistent with the rest of the paper: why not condition on the image alone? A 2D environment with significantly more variations and objects than 5c may suffice to show that it can work in such a more complex setting, though scaling to 3d scenes would be more informative.

(2) The novelty compared to prior works using tree-search in latent space is potentially limited. For example, [15] does use a sampler and a learned collision checker as noted, but removing them essentially reduces that algorithm to ELAST. The state sampler is only used to bias the search and make it more efficient, but also state sampling is likely freely available from your training data. It is not clear to me why one wouldn’t use it. The collision checker is used to allow generalization to novel environments, which in ELAST is handled by the context c. It is not clear how powerfully the context can be used to extend to more complex environments and should be shown by this work.

One major change between 15 and this work is that the latent space is learned in a potentially more robust way through contrastive learning. It would be useful to investigate the effect this has by comparing to VAE-based latent spaces with enforced dynamics.

A few more minor weaknesses:
- An algorithm box for the tree search and the latent space learning would be appreciated.
- Adding the network names to the figures in e.g. Fig 3 and 4 would be helpful to quickly see how each fits together.

Minor notes:
- Line 198: list the inptuts and outputs of h_f.
- ELAST + RL is an interesting extension
- Appendix D1 is interesting and would appreciate more detailed

---

> ### Author Response · Authors · 2022-08-02
> **Answer to Reviewer 6tyY**
>
> We would like to thank the reviewer for the questions and detailed feedback. Our response is divided into (1) complexity of the problems (2) novelty of our approach (3) other comments/questions.
>
> # (1) Complexity of problems
>
> >Robot dimensionality and variation
>
> We appreciate the reviewer's concern that the state/action spaces appear relatively low-dimensional compared, for example, to motion planning benchmarks in classical robotics, where dynamics and symbolic state spaces are known. However, we would like to emphasize that we are addressing planning based on visual observations, a setting for which robust methods are still at an early stage of development. ELAST offers significant performance improvements over existing methods. We evaluate it on planar workspace tasks, but also for control tasks with a 6/7 joint robotic manipulator. The Sawyer robot tasks were taken from Metaworld [1], a popular benchmark for robot learning. We would also like to point out the complexity of the deformable objects tasks (Figure 6. (e) and (f)). In our physics simulation, the cable is modeled by a chain of 43 rigid body segments. One must consider the deformation of the entire cable to (a) navigate around the obstacles and (b) achieve the desired cable deformation shown in the goal image.
>
> >Conditioning on the actual states of parameterization (..) does not seem consistent with the rest of the paper: why not condition on the image alone?"
>
> We believe this is a misunderstanding, possibly caused by an unclear formulation. ELAST conditions on latent encodings obtained by training a convolutional autoencoder on raw context images. We do **not** provide the underlying ground truth poses of the obstacles. We refer to L312/312 "*In parameterized environments, we obtain context observations c by taking the latent encodings given by a convolutional autoencoder which is trained separately on context training images."*.
>
> > A 2D environment with significantly more variations and objects than 5c may suffice to show that it can work in such a more complex setting
>
> > It is not clear how powerfully the context can be used to extend to more complex environments
>
> We agree that more complex parameterizations would be interesting to study the scalability of our method. Therefore, we created a new version of *BlockParam* with more complicated obstacle maps (see new **Appendix D.7**). Each map was generated by randomly selecting the position, orientation, and shape of 8 rectangular obstacles. We demonstrate that ELAST is indeed applicable to such environments with more complicated parameterizations, showing an average success rate of 85%.
>
> # (2) Novelty compared to [2]
>
> As mentioned earlier, [2] trains a collision checker using labeled data. To generate such a dataset, one needs to identify collision states based on the underlying poses and geometries of the objects in the scene. However, this is usually not the case in real-world manipulation scenarios. In [2], a collision checker is required to avoid illegal transitions, and thus is not applicable to the environments in our work. [2] uses an RRT-like strategy, while ours is more similar to EST*. While the basic idea of single-query based planning may be similar, ELAST relies on unsupervised learning techniques to avoid assumptions such as collision checks. Unlike [2], we exploit the properties of our contrastive embedding to rewire the search tree, which significantly improves path length and success rates (see ablation experiments in Appendix D4).
>
> [2] uses a joint embedding for all contexts. In the "RRT" expansion, it only selects states from the dataset ([2] Sec 5,  "By selecting samples only from,.." ). This is to ensure that the search remains on the latent manifold. Thus, the state sampler is not only a heuristic, but becomes necessary. It requires a dataset that sufficiently "covers" the state space+parameterizations. Our approach, on the other hand, leverages context interpolation by learning context-dependent models. ELAST then avoids search outside the latent manifold by rejecting transitions given their predictive uncertainty and conditional density. Compared to [2], we also demonstrate results for larger range and more complex visual tasks (e.g. vision-based manipulator control).
>
> # Other Comments/questions
> > What is the computation time?
>
> We added a section about computation times in the App. D6
>
> >VAE-based latent spaces with enforced dynamics is missing. How robust is the latent space?
>
> We added this comparison to App. D1.
>
> >Why does 19c/d form a loop in the isomap?
>
> We believe the loop exists only in the visualization as a result of ISOMAP trying to map the complex 16dim latent space into a 2d representation. I.e., the corresponding regions might not be overlapping in the original space.
>
> [1] Tianhe et al. Meta-world: A benchmark and evaluation for multi-task and meta reinforcement
> learning. PMLR 20.
>
> [2] Ichter et al. Robot motion planning in learned latent spaces. RA-L 19

---

> > ### Comment · Reviewer_6tyY · 2022-08-08
> > **Thanks for the clarifications and additional experiments**
> >
> > I appreciate the author’s clarifications and additions.
> >
> > The computation times are useful and indeed faster than I expected for the number of iterations. The comparisons to VAE with dynamics are very interesting.
> >
> > The more complex block environment is interesting and nice to see that the performance and embedding perform well with more complex contexts. I think this is crucial to see given the removal of the collision checker and the claims of application to varying context. In terms of novelty, this helps to clarify why the changes for the EST and removal of the collision checker are possible for ELAST.
> >
> > The views of several seeds is really useful as it appears as though the CPC is quite stable. I would be interested to see it for a few additional domains that are already tested. I think this ask is likely outside the scope of the time for rebuttal, but it would be quite interesting to see a domain like D.7 with ~5 to 10 random seeds and the resulting success rate compared to other latent encodings. Showing both isomap and the downstream planning performance is important as the robustness of the latent space is an important contribution of ELAST.
> > > Additional question: Can the authors also confirm that the visualized seeds in 11a are three randomly chosen seeds? The structure is clearly outperforming 11b for instance, but it would be good to have it confirmed how robust the latent space is.
> >
> > Though the authors note that the manipulator and cable have higher dimensional workspaces, the useful workspace is low dimensional (which of course is what the latent encoding discovers and exploits). The arm is mostly in free space and the cable just needs a collision free 2d path, not all dimensions must be considered (or maybe I’m missing something there). I’m loosely defining “useful” workspace as the workspace that must be explored significantly to find a solution. This is why something like even SE(2) or a dubins car (or really ideally something SE(3) or more complex) with some obstacles would be particularly useful. This is an area that can be challenging for latent space planning. Problems inspired by problems here, https://ompl.kavrakilab.org/gallery.html in the section Planning Using OMPL.app would be particularly informative (alpha puzzle, SE(2) bugtrap, Reeds-Shepp maze, SE(3) hole task).
> >
> > From the above, the following would be great for camera ready:
> > - A scene in which the useful workspace is actually higher dimensional
> > - Seeds for latent spaces and their resultant downstream planning performance
> > - Additional seed isomaps for other domains (and confirmation that it's random seeds and not cherry picked)
> > - Adding an algorithm box I think would clarify the approach further still.
> >
> > Based on these I’m amending my score higher.

---

> > > ### Author Response · Authors · 2022-08-08
> > > **Re:Thanks for the clarifications and additional experiments**
> > >
> > > Dear  reviewer,
> > >
> > > Thank you for your response and consideration of the revised version of our paper.
> > >
> > > > Would be quite interesting to see a domain like D.7 with ~5 to 10 random seeds and the resulting success rate compared to other latent encodings. Showing both isomap and the downstream planning performance is important as the robustness of the latent space is an important contribution of ELAST.
> > >
> > > We are happy to provide the Isomap embeddings and corresponding success rates for different random seeds, e.g., for environment D7, in the final version of the paper.
> > >
> > > > Additional question: Can the authors also confirm that the visualized seeds in 11a are three randomly chosen seeds? The structure is clearly outperforming 11b for instance, but it would be good to have it confirmed how robust the latent space is.
> > >
> > > Yes, the visualizations in 11a do indeed show the Isomap embedding for three different randomly selected seeds.
> > >
> > > > Adding an algorithm box I think would clarify the approach further still
> > >
> > > We agree that an algorithm box would be a useful addition to the final version to facilitate understanding of the entire pipeline.
> > >
> > > > E(2) or a dubins car (or really ideally something SE(3) or more complex) with some obstacles would be particularly useful. This is an area that can be challenging for latent space planning. Problems inspired by problems here, https://ompl.kavrakilab.org/gallery.html in the section Planning Using OMPL.app would be particularly informative (alpha puzzle, SE(2) bugtrap, Reeds-Shepp maze, SE(3) hole task).
> > >
> > > We thank you for your suggestions. We will consider additional experiments. A Dubins car might be interesting to investigate feasibility under clearly non-holonomic kinematic constraints. However, a general challenge w.r.t. the suggested environments might be how to fully capture the robot’s states when we only have raw visual observations. One solution could be to have access to proprioceptive sensors (a common assumption in robotics) and use this information, along with latent encodings of the image data, for planning with ELAST. This strategy would likely allow scaling to more complex environments and robotic systems.

---

### Author Response · Authors · 2022-08-07
**General response**

Dear reviewers,

Thank you for reading our work and evaluating it so thoroughly. We hope that we have answered all your questions and cleared up any misunderstandings. We would like to emphasize again that we updated the manuscript and provided additional experimental results, such as computation times and evaluation under more complex obstacles, as requested by some reviewers (this was submitted simultaneously with the author response). We are ready to participate in a discussion and look forward to answering any open questions that may remain.

Sincerely,
The authors

---

### Meta-Review · Area_Chair_4pK4 · 2022-08-23

**Recommendation:** Accept
**Confidence:** Certain

**Metareview:**

This work proposes a method to first learn a lower dimensional embedding via contrastive learning, then learns a transition model and then utilizes a planner inspired from sampling-based motion planner literature to plan in this latent space from start to goal states. A model-predictive controller is harnessed to follow the desired trajectories in latent space. Overall, the work is well presented and has been well received by reviewers. It should inspire techniques in a lot of related areas like vision-language navigation. During author-reviewer discussion period there was rich interaction and various additional clarifications, and experiments were added by the authors. The authors are encouraged to incorporate them into camera-ready version and release reproducible source-code to accompany the paper.

**Award:**

No

---

### Decision · Program_Chairs · 2022-09-14

Accept